# ARPC5 isoforms and their regulation by calcium-calmodulin-N-WASP drive distinct Arp2/3-dependent actin remodeling events in CD4 T cells

Lopamudra Sadhu[1], Nikolaos Tsopoulidis[1]*[†], Md Hasanuzzaman[1], Vibor Laketa[2], Michael Way[3,4], Oliver T Fackler[1]*

[1]Department of Infectious Diseases, Integrative Virology, University Hospital Heidelberg, Heidelberg, Germany; [2]Department of Infectious Diseases, Virology, University Hospital Heidelberg, Heidelberg, Germany; [3]Cellular Signalling and Cytoskeletal Function Laboratory, The Francis Crick Institute, London, United Kingdom; [4]Department of Infectious Disease, Imperial College, London, United Kingdom

*For correspondence:
ntsopoul@gmail.com (NT);
oliver.fackler@med.uni-heidelberg.de (OTF)

Present address: [†]Department of Molecular Biology, Mass General Hospital, Boston, United States

Competing interest: The authors declare that no competing interests exist.

**Abstract** CD4 T cell activation induces nuclear and cytoplasmic actin polymerization via the Arp2/3 complex to activate cytokine expression and strengthen T cell receptor (TCR) signaling. Actin polymerization dynamics and filament morphology differ between nucleus and cytoplasm. However, it is unclear how the Arp2/3 complex mediates distinct nuclear and cytoplasmic actin polymerization in response to a common stimulus. In humans, the ARP3, ARPC1, and ARPC5 subunits of the Arp2/3 complex exist as two different isoforms, resulting in complexes with different properties. Here, we show that the Arp2/3 subunit isoforms ARPC5 and ARPC5L play a central role in coordinating distinct actin polymerization events in CD4 T cells. While ARPC5L is heterogeneously expressed in individual CD4 T cells, it specifically drives nuclear actin polymerization upon T cell activation. In contrast, ARPC5 is evenly expressed in CD4 T cell populations and is required for cytoplasmic actin dynamics. Interestingly, nuclear actin polymerization triggered by a different stimulus, DNA replication stress, specifically requires ARPC5 but not ARPC5L. TCR signaling but not DNA replication stress induces nuclear actin polymerization via nuclear calcium-calmodulin signaling and N-WASP. Diversity in the molecular properties and individual expression patterns of ARPC5 subunit isoforms thus tailors Arp2/3-mediated actin polymerization to different physiological stimuli.

## Editor's evaluation

This fundamental study demonstrates that the two isoforms of the ARPC5 subunit (ARPC5 and ARPC5L) of the Arp2/3 complex have specific functions in regulating cytoplasmic and nuclear actin filament assembly in response to DNA replication stress and T cell receptor signaling in T lymphocytes. The data presented in the manuscript are convincing and of good technical quality, and the study provides interesting new insights into specific cellular roles of different Arp2/3 isoforms in T lymphocytes.

## Introduction

Development, proliferation, and immune functions of T lymphocytes are regulated by their activation state. In concert with co-stimulatory receptors such as CD28, T cell activation is primarily governed by

engagement of surface exposed T cell antigen receptor (TCR/CD3) complexes with major histocompatibility complex (MHC) II-bound peptides on antigen-presenting cells (APCs) (*Dustin, 2008*; *Grakoui et al., 1999*; *Sedwick et al., 1999*; *Tskvitaria-Fuller et al., 2003*; *Valitutti et al., 1995*). T cell activation triggers proliferation and polarized differentiation of naïve T cells required for their development into effector T cells (*Constant et al., 1995*; *Kaech et al., 2002*; *Zhu and Paul, 2008*). Physiologically, APC-T cell interactions occur in the context of stable cell–cell contacts referred to as immunological synapses and trigger a broad range of downstream signaling events, including a sequential tyrosine phosphorylation cascades to activate PKC and MAPK signaling, as well as rapid elevation of intracellular $Ca^{2+}$ levels, which together regulates the expression of TCR target genes (*Arendt et al., 2002*; *Dustin, 2008*; *Joseph et al., 2014*; *Monaco et al., 2016*; *Monks et al., 1997*; *Oh-hora and Rao, 2008*; *Rhee and Choi, 1992*; *Tsopoulidis et al., 2019*). TCR engagement also triggers the immediate polymerization of cortical actin at the immune synapse, which regulates downstream signaling by facilitating the formation and proper spatial distribution of signaling competent microclusters as well as by coordinating TCR and integrin signaling (*Billadeau et al., 2007*; *Jankowska et al., 2018*; *Morimoto et al., 2000*; *Valitutti et al., 1995*; *Varma et al., 2006*).

In addition to the cytoplasm, actin is also highly abundant in the cell nucleus, but the role of nuclear actin is much less studied or understood. The advent of improved probes to visualize nuclear F-actin facilitated the detection of nuclear actin assembly into complex structures (*Baarlink et al., 2013*; *Melak et al., 2017*). Following the initial observation of short-lived F-actin networks in fibroblasts upon stimulation with serum, evidence for the formation of nuclear actin filaments in mammalian cells, for example, upon integrin and G-protein-coupled receptor signaling or DNA damage response and chromosome break repair, DNA replication stress, postmitotic nucleus expansion, or infection with cytomegalovirus emerged (*Baarlink et al., 2017*; *Baarlink et al., 2013*; *Belin et al., 2013*; *Caridi et al., 2018*; *Lamm et al., 2020*; *Schrank et al., 2018*; *Wang et al., 2019*; *Wilkie et al., 2016*). The morphology of nuclear actin filaments ranges from stable thick cables to very transient thinner filament bundles. Different actin nucleators have been implicated in the formation of the nuclear actin filaments: the formins Dia1 or Formin-1/Spire1/2 for F-actin networks triggered by serum and integrin signaling or DNA damage, or Arp2/3 complex for nuclear actin filaments induced in response to DNA damage (*Baarlink et al., 2013*; *Belin et al., 2015*; *Caridi et al., 2018*; *Lamm et al., 2020*; *Wang et al., 2019*).

Analyzing nuclear actin dynamics in response to TCR engagement in CD4 T cells, we previously observed the rapid induction of a transient meshwork of thin nuclear actin filaments within seconds after stimulation that is subsequently disassembled after a couple of minutes (*Tsopoulidis et al., 2019*). This nuclear actin assembly, which drives a selective gene expression program required for the helper function of CD4 T cells, is triggered by nuclear calcium–calmodulin signaling and precedes actin polymerization in the cytoplasm. Interestingly, TCR-induced actin polymerization in the nucleus as well as in the cytoplasm relies on the Arp2/3 complex, but how spatiotemporal control of these distinct actin polymerization events is achieved is unclear. Arp2/3 is a seven subunit complex that mediates actin polymerization in a wide range of diverse cellular processes, including the formation of lamellipodia, endocytosis, and/or phagocytosis at the plasma membrane. Further, Arp2/3 activity is regulated by interaction with nucleation-promoting factors (NPFs), including WASP, N-WASP, WASH, and WAVE (*Pizarro-Cerdá et al., 2017*; *Rottner et al., 2010*). NPF binding to and concomitant activation of Arp2/3 is induced by diverse upstream signals, and the involvement of different NPFs therefore represents one level of spatiotemporal control of the Arp2/3 complex. In addition, the Arp2/3 subunits ARP3, ARPC1, and ARPC5 exist as two different isoforms in humans that can assemble into complexes with different biochemical properties (*Abella et al., 2016*; *Balasubramanian et al., 1996*; *Galloni et al., 2021*; *Jay et al., 2000*; *Millard et al., 2003*; *von Loeffelholz et al., 2020*). As recently demonstrated for ARP3, different subunit isoforms can provide the Arp2/3 complex with sensitivity for distinct upstream regulation (*Galloni et al., 2021*). To assess how Arp2/3 can mediate distinct actin polymerization events in the nucleus and cytoplasm in response to a common stimulus, we investigated herein the role of Arp2/3 subunit isoforms and NPFs in these processes.

## Results

### Arp2/3 regulates nuclear and cytoplasmic actin polymerization during CD4 T cell activation

Transient nuclear and cytoplasmic actin remodeling in response to T cell activation can be visualized using several experimental approaches, including live-cell imaging using Jurkat CD4 T cells stably expressing a nuclear lifeact-GFP reporter (JNLA). Importantly, the endogenous nuclear F-actin meshwork can also be detected by F-actin staining of stimulated CD4 T cells that did not express any reporter (*Tsopoulidis et al., 2019*). Immune synapses between T cells and APCs can be induced via superantigens like Staphylococcus enterotoxin E (SEE), that cross-links unspecifically MHCII with the TCR and results in potent T cell activation. Upon immune synapse formation between JNLA cells with SEE-loaded Raji B cells (*Figure 1A–D*, *Video 1*), actin polymerization is first observed in the nucleus and is followed by actin polymerization at the cell–cell contact site. These processes are recapitulated by surface-mediated stimulation of JNLA cells when plated on dishes coated with anti-CD3 and anti-CD28 antibodies (*Figure 1E*): TCR engagement rapidly induced a nuclear F-actin (NFA) meshwork, followed by cell spreading and actin polymerization into a circumferential F-actin ring (AR) at the cell periphery (*Figure 1F and G*, *Video 2*) followed by subsequent disassembly of actin filaments. The simultaneous visualization of nuclear and cytoplasmic actin dynamics is challenging due to their transient nature and occurrence in different focal planes. To specifically study nuclear actin polymerization, we therefore often induce T cell activation by PMA/ionomycin (P/I), which only triggers nuclear actin polymerization in the absence of cytoplasmic actin remodeling or cell spreading. The formation of nuclear F-actin can be triggered by ionomycin alone, but for optimal nuclear actin polymerization, ionomycin is typically used in combination with PMA (*Tsopoulidis et al., 2019*; *Figure 1H–J*). Generally, the formation of both NFA and AR can be blocked with comparable efficacy by pretreating the cells with the Arp2/3 inhibitor CK869, regardless of the stimulus type (*Figure 1C, D, G and J* and *Video 3*, *Video 4*). Within the short time frame of the experiment, CK869 treatment does not cause significant changes in CD3 surface expression, excluding receptor downregulation as cause for reduced actin polymerization upon stimulation with anti-CD3 antibody (*Figure 1—figure supplement 1*). Arp2/3 complex thus mediates distinct phenotypically discernable actin polymerization events in CD4 T cell activation that are equally sensitive to pharmacological interference independently of the T cell stimulus used.

### Heterogeneous expression of ARP2/3 isoforms in CD4 T cells

Nuclear actin polymerization following T cell activation depends on nuclear $Ca^{2+}$ transients, while cytoplasmic actin polymerization is largely independent from $Ca^{2+}$ signaling. However, both nuclear and cytoplasmic actin polymerization require activation of the Arp2/3 complex and only a subset of T cells undergoes nuclear actin polymerization, that is, approximately 25% Jurkat T cells and 65% of primary CD4 T cells display nuclear actin polymerization upon stimulation (*Figure 1*; *Tsopoulidis et al., 2019*). These findings suggest that differences in the nuclear actin polymerization response of CD4 T cell populations may be governed by molecular diversity. In search for the molecular basis for this differential regulation of the Arp2/3 complex, we hypothesized that nuclear and cytoplasmic actin polymerization might involve distinct Arp2/3-complex subunit isoforms (*Abella et al., 2016*; *Millard et al., 2003*). Analyzing the protein expression profile of Arp2/3 subunit isoforms in Jurkat CD4 T cells and primary human CD4 T cells revealed expression of all Arp2/3 complex subunits (*Figure 2A*); detection of ARPC5, however, required extended exposure. Primary resting CD4 T cells displayed low expression levels of all Arp2/3 complex subunits, but T cell activation increased their levels significantly. This induction of protein levels by T cell activation was paralleled by induction of mRNA expression assessed by qRT-PCR (*Figure 2—figure supplement 1A*: absolute values; *Figure 2—figure supplement 1B*: relative to GAPDH; note that equal number of cells was loaded and that housekeeping genes are also subject to regulation by T cell activation; *Roy et al., 2020*; *Sousa et al., 2019*; *Subbannayya et al., 2020*).

We noted that baseline mRNA expression of ARPC1A and ARPC5L was consistently lower in bulk Jurkat and primary CD4 T cell cultures compared to other subunit isoforms and was not strongly increased upon T cell activation (*Figure 2—figure supplement 1A and B*). The consistently low expression of ARPC1A and ARPC5L could result from weak expression of these isoforms in every cell or a heterogeneous expression pattern that manifests as overall low expression levels in bulk cultures.

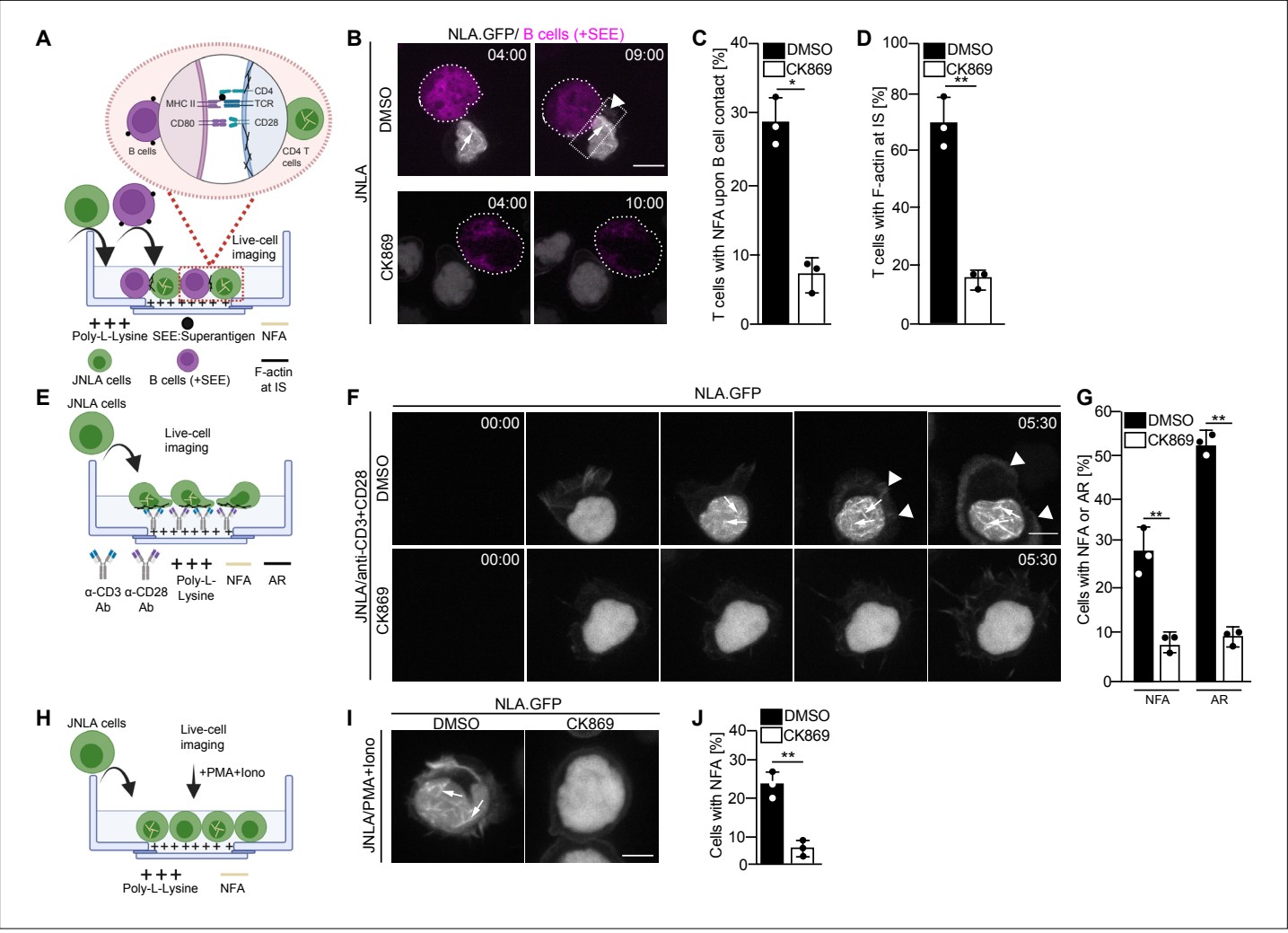

**Figure 1.** Arp2/3 complex-mediated nuclear and plasma membrane actin polymerization in CD4 T cells. (**A**) Schematic representation of the experimental setup to visualize actin dynamics at the immune synapse, as performed for **B–D**. (**B**) Shown are representative still images at indicated time points from live-cell visualization of nuclear and plasma membrane actin dynamics in T cells stably expressing nuclear lifeact-GFP (referred to as JNLA) treated with either DMSO (solvent control) or CK869 upon contact with Staphylococcus enterotoxin E (SEE) pulsed Raji B cells. Images were acquired every 70 s for a total of 30 min after adding the Raji B cells. Still images represent the time at which the T and B cells made contact (left panel) to the time they formed an immune synapse (IS) as shown by the accumulation of plasma membrane F-actin at the contact site (right panel). (**C**) Quantification of nuclear (NFA) and (**D**) plasma membrane (AR) F-actin dynamics of JNLA cells upon contact with SEE pulsed Raji B cells is shown, respectively. All data points indicate mean ± SD values from three independent experiments with at least 40 cells analyzed per condition per experiment. Scale bar, 7 μm. (**E**) Schematic representation of the experimental/live-cell imaging setup on stimulatory GBDs as performed for **F–G**. (**F**) JNLA pretreated with either DMSO (solvent control) or CK869 for 30 min were put on TCR stimulatory GBDs and subjected to live-cell microscopy. Shown are representative still images from the spinning-disk confocal microscope from the time the cells fall on the coverslips until after contact with the stimulatory surface, with acquisition every 30 s. Arrows indicate the nuclear F-actin (NFA), whereas arrowheads point to the F-actin at PM. Quantification of (**G**) nuclear actin filaments (NFA) and plasma membrane F-actin ring (AR) polymerization is shown, respectively, upon contact with TCR stimulatory surface. Data points indicate mean ± SD values from three independent experiments where 40–60 cells were analyzed per condition in each experiment. (**H**) Schematic representation of the experimental/live cell imaging setup with P/I activation as performed for **I** and **J**. Relates to *Figure 1—figure supplement 1A–E*. (**I**) JNLA cells pretreated with either DMSO (solvent control) or CK869 for 30 min were put on poly-lysine-coated GBDs and subjected to live-cell microscopy, which was then followed by addition of P/I. Shown are representative still images from the spinning-disk confocal microscope forming NFA after P/I addition. Arrows indicate the NFA. (**J**) Quantification of nuclear actin polymerization upon addition of P/I was performed. In (**C**, **D**, **G**, **J**), each data point indicates the mean value of an independent experiment with 40–60 cells analyzed per condition with indicated mean ± SD from three independent experiments. Scale bar, 3 μm. NFA is denoted as yellow filaments within nucleus, whereas plasma membrane f-actin is denoted as black filaments across all experimental schematics shown. Statistical significance based on the calculation of mean ± SD from three independent experiments using Welch's *t*-test was performed. *p≤0.0332, **p≤0.0021, and ns: not significant. Source data is avaialble at https://doi.org/10.11588/data/YVYEO8.

The online version of this article includes the following figure supplement(s) for figure 1:

**Figure supplement 1.** CD3 expression in CK869-treated cells upon PMA + ionomycin (PI) stimulation.

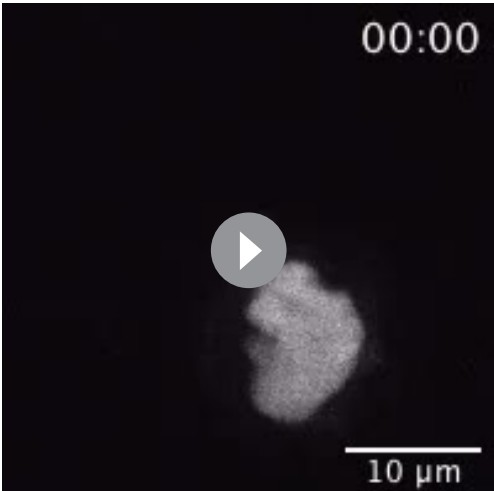

**Video 1.** Live imaging of an immune synapse (IS) formation between DMSO-treated JNLA (in gray) and Staphylococcus enterotoxin E (SEE)-treated B cells (in magenta).

https://elifesciences.org/articles/82450/figures#video1

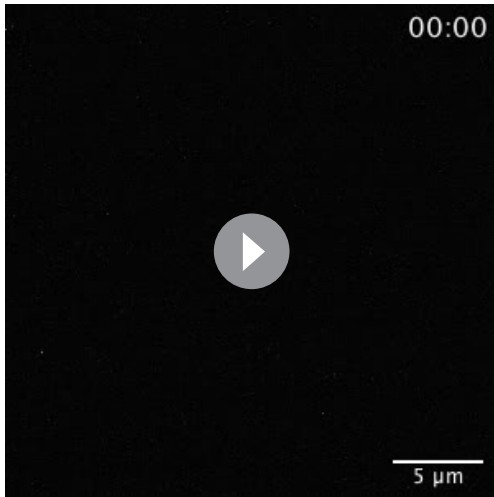

**Video 2.** Live nuclear F-actin (NFA) and F-actin ring (AR) formation in DMSO-treated Jurkat CD4 T cells upon falling on a stimulatory surface.

https://elifesciences.org/articles/82450/figures#video2

To assess whether individual cells display heterogeneous expression of Arp2/3 subunit isoforms, we performed single-cell RNA-sequencing (scRNA-Seq) analysis of publicly available CD4 T cell data sets. As expected, UMAP clustering of Jurkat T cells revealed an overall homogenous cell population (*Figure 2B*). Quantification of Arp2/3 subunit isoform expression levels showed robust expression of ARP2, ARP3, and ARPC5, an intermediate level of expression of ARPC1B, and relatively low expression of ARPC1A and ARPC5L, consistent with the findings from the bulk analysis. To quantify the number of cells robustly expressing these isoforms, we counted all cells that expressed a given gene above the cutoff level of 1 transcript per million (TPM). Interestingly, ARPC1A and ARPC5L were strongly expressed in only 33 and 46% of Jurkat cells, respectively. Since ARPC1A and ARPC5L expression was not associated with a particular cell-cycle phase (*Figure 2—figure supplement 1E*), we assume that these two proteins are

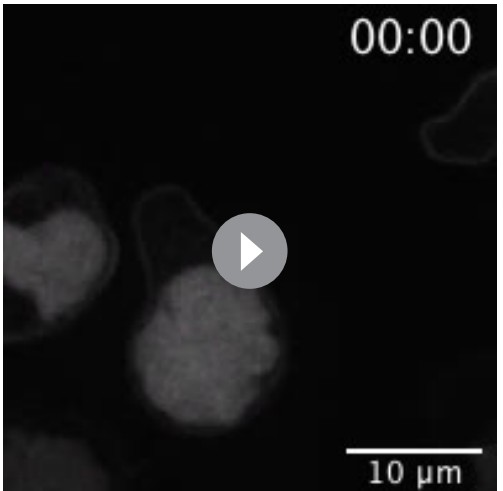

**Video 3.** Live imaging of an immune synapse (IS) formation between CK869-treated JNLA (in gray) and Staphylococcus enterotoxin E (SEE)-treated B cells (in magenta).

https://elifesciences.org/articles/82450/figures#video3

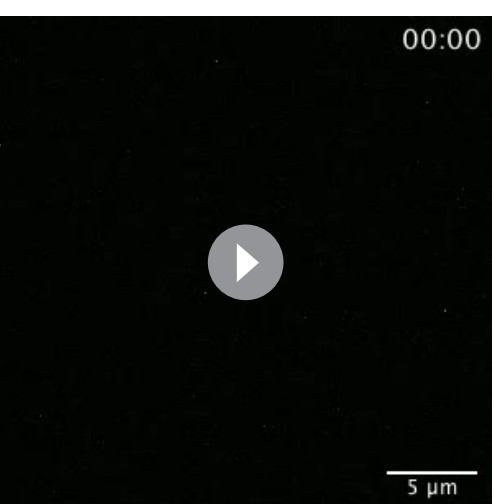

**Video 4.** Live nuclear F-actin (NFA) and F-actin ring (AR) formation in CK869-treated Jurkat CD4 T cells upon falling on a stimulatory surface.

https://elifesciences.org/articles/82450/figures#video4

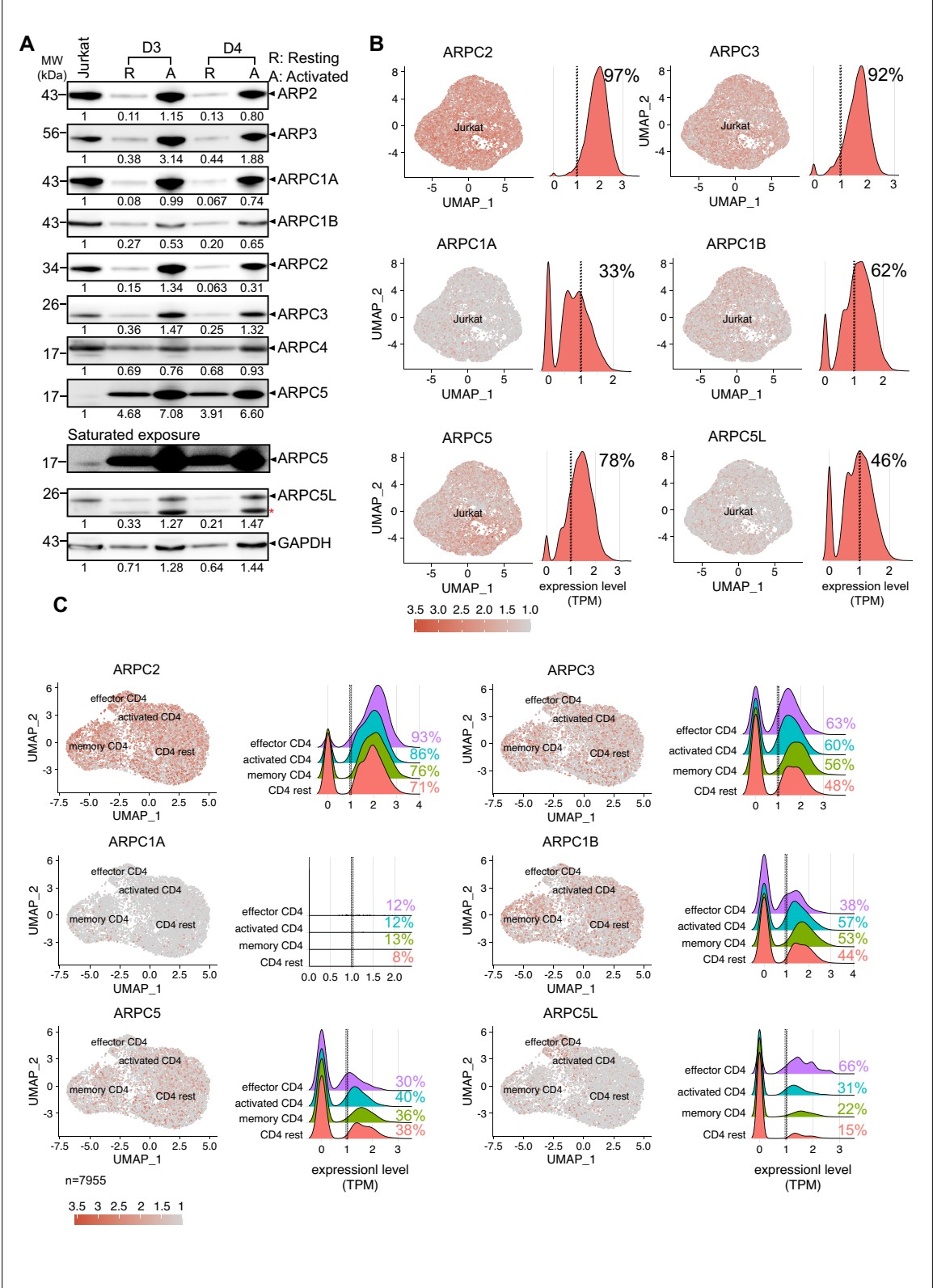

**Figure 2.** Heterogeneous expression of ARP2/3 isoforms in CD4 T cells. (**A**) Expression of all the subunits of the Arp2/3 complex along with the respective isoforms of ARPC1 and ARPC5 across Jurkat cell line and primary human CD4 T cells from two representative healthy donors were verified using western blotting. Representative immunoblots compare the protein levels of each subunit and their isoforms in CD4 T cells. Additional comparisons for expression of these proteins in Resting (R) and Activated (A) CD4 T cells from donors 3 and 4 are shown, respectively. Black arrowheads

*Figure 2 continued on next page*

*Figure 2 continued*

indicate the specific bands. Note that the ARPC5L antibody also detects ARPC5 (marked by red asterisk). The numbers indicated below each row represents the mean ± SD values from three independent experiments of the densitometric quantification of the bands compared to Jurkat protein levels (which is set to 1). Saturated exposure of the ARPC5 immunoblot is shown for better visualization of the ARPC5 levels in the JNLA cell line. (**B, C**) Single-cell RNA-sequencing analysis of Jurkat CD4 T cells (**B**) and primary CD4 T cells (**C**). Expression of selected genes in UMAP embedded cells with adjacent histograms of their frequency distributions. See *Figure 2—figure supplement 1C* for UMAP embedding of (**C**). Dashed line represents threshold for frequency quantification at 1 TPM. Source data is avaialble at https://doi.org/10.11588/data/YVYEO8.

The online version of this article includes the following figure supplement(s) for figure 2:

**Figure supplement 1.** Heterogeneous expression of ARP2/3 isoforms in CD4 T cells.

heterogeneously expressed within Jurkat cells, where some cells show robust expression, and the majority exhibits weak or undetectable expression.

Next, we wondered whether this heterogeneity is also represented in primary human CD4 T cells, which are inherently diverse in nature. UMAP embedding of scRNA-Seq data from lymph node-derived CD4 T cells (*Szabo et al., 2019*) allowed us to define four cellular clusters based on conventional markers: resting CD4, memory CD4, activated CD4, and effector T cells (*Figure 2—figure supplement 1C and D*). While ARPC3, ARPC2, ARPC1B, and ARPC5 were strongly and homogenously expressed throughout the clusters, ARPC1A was very weakly expressed and only barely detectable (*Figure 2C*). Interestingly, the highest heterogeneity of expression was displayed by ARPC5L, and its expression levels correlated with the activation status of CD4 T cells, ranging from 15% in resting CD4 T cells to 22% and 31% in memory and activated CD4 T cells to 66% in effector CD4 T cells. Based on this observation, it appears that there is a functional relationship between T cell activation and expression of ARPC5L isoform containing Arp2/3-complexes. In sum, all Arp2/3 subunit isoforms are expressed in Jurkat and primary CD4 T cells, but their expression is heterogeneous in individual cells. Of all subunits, ARPC5L displays the highest heterogenic expression within a given T cell population and the frequency of cells that express ARPC5L is similar to that of cells displaying nuclear actin polymerization in response to T cell activation. Heterogeneously expressed molecules may thus be molecular discriminators that govern nuclear actin polymerization in CD4 T cells.

## Distinct ARPC5 isoforms mediate cytoplasmic and nuclear actin polymerization induced by TCR signaling

Motivated by the notion that Arp2/3-complex isoform subunits might be discriminators that govern actin polymerization in T cells, we went on to systematically test the role of these Arp2/3 subunit isoforms in the context of nuclear and cytoplasmic actin polymerization triggered by TCR signaling. First, we sought to selectively suppress expression of individual isoforms. However, combining gene silencing with expression of a nuclear lifeact-GFP reporter while maintaining a low activation state that would allow us to study the response to T cell activation in primary T cells proved to be challenging. We therefore focused all subsequent work on cell line models, which faithfully capture many aspects of T cell activation-induced actin remodeling including the heterogeneity of Arp2/3 complex subunit expression (*Figure 2B*). For this, we transduced bulk JNLA cultures with isoform-specific shRNAs to reduce the expression of ARPC1A, ARPC1B, ARPC5, or ARPC5L (*Figure 3A*). Selective silencing of ARPC1A, ARPC1B, or ARPC5 did not significantly reduce the frequency of cells that displayed NFA in response to P/I. In contrast, JNLA cells with reduced ARPC5L levels were significantly impaired in NFA formation upon stimulation with P/I but also in response to TCR engagement (*Figure 3B and C*, *Figure 3—figure supplement 1A and B*). As with NFA, ARPC1A or ARPC1B silenced cells were both able to support cell spreading and formation of cytoplasmic AR after surface-mediated TCR stimulation (*Figure 3D and E*, see *Figure 3—figure supplement 1C* for a lower magnification overview and *Figure 3—figure supplement 1D–F* for quantification of cell morphologies). Importantly, ARPC5 but not ARPC5L was required for efficient cell spreading and cytoplasmic AR assembly triggered by TCR engagement and cells lacking ARPC5 often displayed aberrant F-actin organization (*Figure 3—figure supplement 1C, E, and F*).

## Knockout and reconstitution of ARPC5 isoforms

Our observations pointed to a central role for ARPC5 and ARPC5L in the specificity of Arp2/3-driven cytoplasmic and nuclear actin polymerization in response to CD4 T cell activation. To validate our

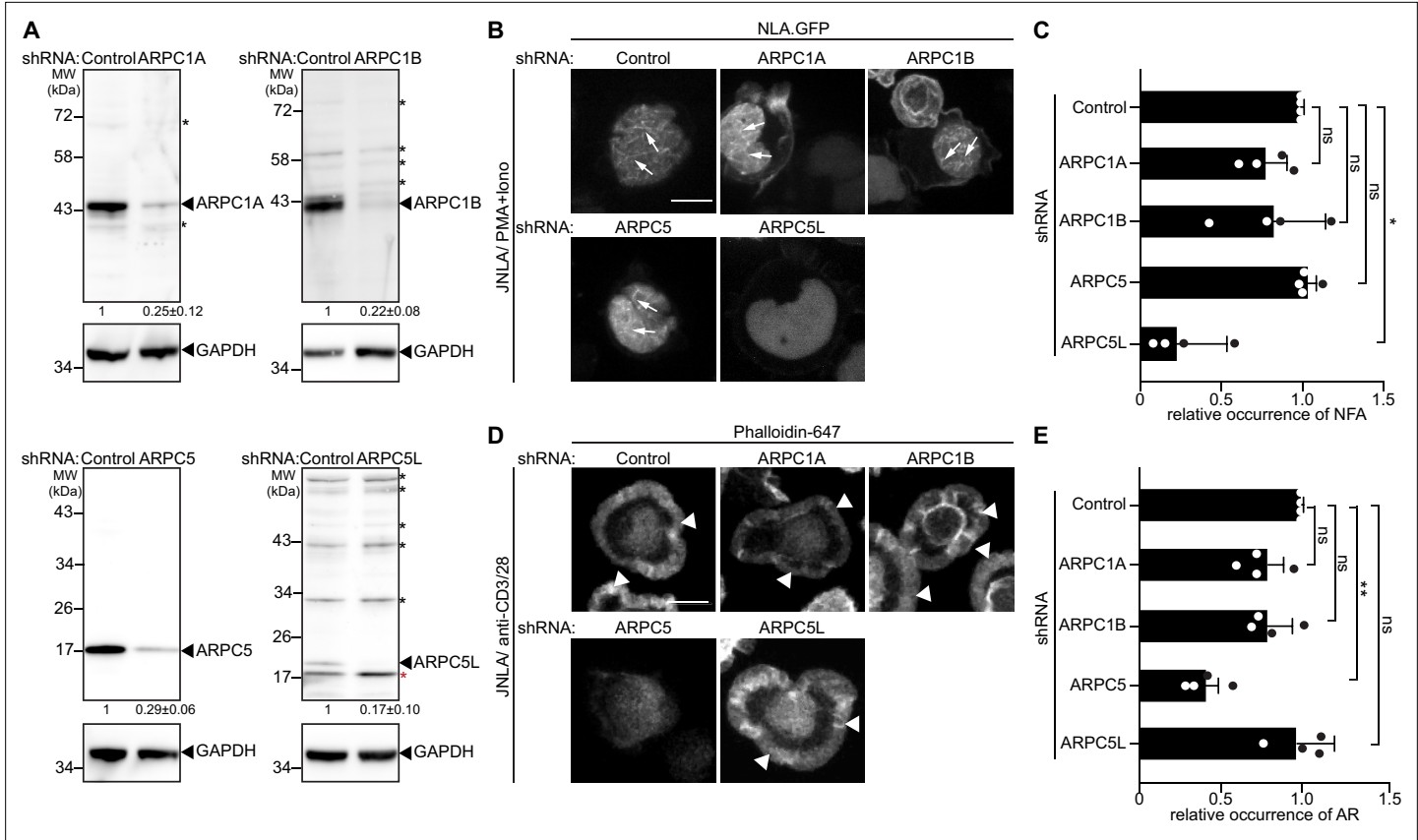

**Figure 3.** ARPC5 isoforms differentially regulate nuclear and plasma membrane actin polymerization. (**A**) Representative immunoblots show knockdown of ARPC1 and ARPC5 isoforms in bulk JNLA cells treated with indicated shRNAs. Black arrowheads indicate the specific bands, and black asterisks mark unspecific bands. Note that the ARPC5L antibody also detects ARPC5 (marked by red asterisk). The numbers indicated below the respective blots represent the mean ± SD values from four independent experiments, based on the densitometric quantification of the bands, normalized to GAPDH and compared to the non-targeting control (NTC) protein levels (which is set to 1). (**B**) Representative spinning-disk confocal still images of JNLA cells treated with indicated shRNA show stills post activation with PMA/ionomycin (P/I). Arrows point to the nuclear F-actin (NFA). Relates to *Figure 3—figure supplement 1A,B*. Scale bar, 3 µm. (**C**) Quantification of NFA formation in shRNA-treated cells relative to the scrambled control-treated cells. Mean ± SD of four independent experiments where 30 cells were analyzed per condition per experiment. Each dot represents the mean of each independent experiment. (**D**) Representative immunofluorescence images indicate averaged intensity projections of Phalloidin-647-stained F-actin ring (AR) formation in JNLA cells treated with indicated shRNA upon activation on coverslips coated with anti-CD3+CD28 antibodies. Arrowheads point to the F-actin ring at the PM. (**E**) Quantification of Phalloidin-stained AR formation in shRNA-treated cells relative to the control-treated cells. In (**C**, **E**), each data point indicates the mean value of an independent experiment consisting of two technical replicates with at least 100 cells analyzed per condition with indicated mean ± SD from four independent experiments. One-way ANOVA with Kruskal–Wallis test was used to determine statistical significances, where *$p \leq 0.0332$, **$p \leq 0.0021$, and ns: not significant. Scale bar, 5 µm. See also related *Figure 3—figure supplement 1C-F*. Source data is avaialble at https://doi.org/10.11588/data/YVYEO8.

The online version of this article includes the following figure supplement(s) for figure 3:

**Figure supplement 1.** ARPC5 knockdown cells exhibit filopodia and lamellipodia-like morphotypes upon TCR activation.

observations with transient silencing, we generated ARPC5 or ARPC5L knockout (KO) cell lines using CRISPR-Cas9 ribonucleoprotein transfection. In addition, we reintroduced transgenic ARPC5 or ARPC5L mCherry fusions to rescue the KO or mCherry alone as control (*Figure 4*). Endogenous ARPC5 or ARPC5L levels were strongly reduced (by at least 80%) in the resulting bulk KO cultures as well as in clones expanded from single KO cells (*Figure 4—figure supplements 1A and 2A*). The levels of other Arp2/3 subunits were mostly unaffected, but the expression of ARPC1B was reduced by almost twofold compared to the non-targeting control (NTC) in both KO cultures. ARPC5 KO, on the other hand, led to a decrease in ARPC1A levels (to 76%) and an increase in ARPC5L levels (to 240%) (*Figure 4—figure supplement 1A*). The nucleofection and transduction procedures used to generate and study these KO cells slightly reduced the overall efficiency of NFA formation in

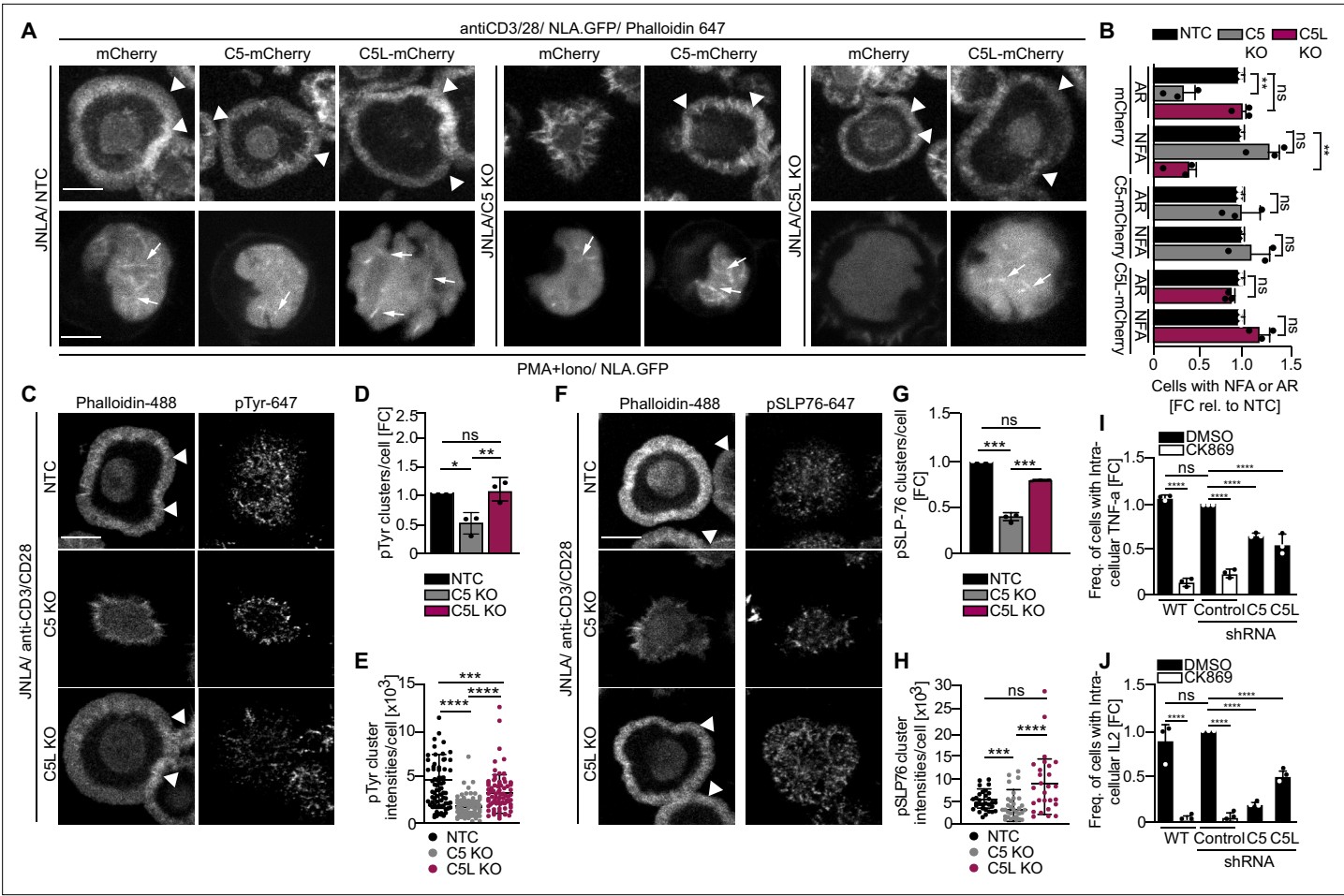

**Figure 4.** Effects observed on nuclear and plasma membrane F-actin dynamics upon ARPC5/C5L knockout and its impact on proximal TCR signaling and cytokine production. (**A**) Shown are representative maximum intensity projections of confocal still images of the indicated KO JNLA cells overexpressing mCherry (control) or mCherry fusion proteins of the respective ARPC5 isoforms, post activation with either anti-CD3+CD28 antibodies (top panel) or with PMA/ionomycin (P/I) (bottom panel). Arrows point to the nuclear F-actin (NFA, bottom). Scale bar, 3 μm. Arrowheads point to the F-actin ring (top). Scale bar, 5 μm. See related *Figure 4—figure supplement 1A-D* and *Figure 4—figure supplement 2A* for western blots of KO and overexpression and *Figure 4—figure supplement 2B and D* for NFA data. (**B**) Quantification of F-actin ring (AR) formation in the PM, stained with Phalloidin-647, is compared to the NFA formation visualized with NLA-GFP in the respective KO or KO+ARPC5 isoform expressing cells was performed relative to the non-targeting control (NTC)-treated cells. 'mCherry' alone was used as vector backbone control for the overexpression study. Bars indicate mean from three independent experiment where 30–40 cells was analyzed per condition. One-way ANOVA with Kruskal–Wallis test was used to determine statistical significances, where **$p \leq 0.0021$ and ns: not significant. See related *Figure 4—figure supplement 2C,E*. (**C, F**) Representative confocal images of JNLA.GFP cells with indicated knockout or control (NTC) upon 5 min of activation on coverslips coated with anti-CD3+CD28 antibodies. Cells were fixed and stained for F-actin (with Phalloidin 488) and pTyr or pSLP-76 (Alexa Fluor 647), respectively. (**D and G**) show quantification of the total number of pTyr or pSLP-76 clusters/cell in KO condition relative to control cells analyzed using the 'Spot Detector' Fiji plugin. Data presented here are mean ± SD from three independent experiments in which two technical replicates were measured per sample. (**E, H**) Dot plots represent the changes in overall intensity of pTyr or pSLP-76 clusters per cell where each dot represents intensity of clusters/cell analyzed manually using Fiji. In (**B**, **D**, **E**), each data point indicates the mean value of an independent experiments with at least 80 cells analyzed per condition with the indicated mean ± SD from three independent experiments. One-sample *t*-test was used to determine statistical significances, where *$p \leq 0.033$, **$p \leq 0.0021$, ***$p \leq 0.0002$ and ns: not significant. Scale bar, 5 μm. (**I, J**) Cytokine production of A30.1 cells treated with the indicated shRNAs in response to P/I. Bars show intracellular levels of TNFα or IL-2, 4 hr or 16 hr of post activation relative to cells treated with the scrambled shRNA control. One-way ANOVA with Tukey's multiple-comparison test was used to determine statistical significances ( **$p \leq 0.005$, ***$p \leq 0.0002$, ****$p \leq 0.0001$, and ns: not significant). Relates to *Figure 4—figure supplement 3A–E*.

The online version of this article includes the following figure supplement(s) for figure 4:

**Figure supplement 1.** Expression of Arp2/3 subunits upon knockout (KO) of respective ARPC5 isoforms and validation of the overexpression of C5 isoforms in the bulk culture.

*Figure 4 continued on next page*

*Figure 4 continued*

**Figure supplement 2.** Rescue of the nuclear F-actin (NFA) and F-actin ring (AR) upon overexpression of the ARPC5 isoforms in JNLA knockout (KO) expanded from single KO clones.

**Figure supplement 3.** Intracellular staining and detection of TNFa and IL-2 in A.301 cells with ARPC5 isoform knockdown.

response to T cell activation. Nevertheless, T cell stimulation confirmed that ARPC5 is selectively required for cytoplasmic actin polymerization and is not substituted by the elevated levels of ARPC5L. In turn, ARPC5L is essential for NFA formation and dispensable for cytoplasmic actin polymerization (*Figure 4A and B*, see mCherry controls, and *Figure 4—figure supplement 2B–E*). Reintroduction of the respective mCherry-tagged ARPC5 isoform in these bulk (*Figure 4A and B*, *Figure 4—figure supplement 1B and C*) and clonal (*Figure 4—figure supplement 2A–E*) KO cells reconstituted their ability to form NFA or ARs, respectively, at expression levels that ranged from significantly lower levels than the endogenous protein to significant overexpression (*Figure 4—figure supplement 1B and C*, *Figure 4—figure supplement 2A*). As previously shown in HeLa cells (*Abella et al., 2016*), immuno-isolation of ectopically expressed ARPC5 isoforms tagged with GFP indicated that they were incorporated into Arp2/3 complexes. Similarly, we confirmed mCherry-tagged ARPC5 isoforms to associate with Arp2/3 with similar efficacy in CD4 T cells (*Figure 4—figure supplement 1D*). ARPC5 and ARPC5L are thus involved in distinct Arp2/3-dependent actin polymerization events during CD4 T cell activation.

## ARPC5L is dispensable for TCR proximal signaling but contributes to effector cytokine expression

Actin polymerization at sites of TCR engagement is directly coupled to downstream signaling constituted by dynamic phosphorylation cascades occurring in microclusters (*Dustin et al., 2006*; *Grakoui et al., 1999*; *Monks et al., 1998*). ARPC5L-driven nuclear actin dynamics precede ARPC5-guided cytoplasmic actin polymerization upon TCR stimulation but whether ARPC5L affects microcluster formation at the plasma mebrane or their function is unclear. We therefore tested whether ARPC5 isoforms differently impact generation and composition of these microclusters formed in response to surface-mediated TCR engagement. Disruption of cytoplasmic actin polymerization upon KO of ARPC5 significantly reduced the number of signaling microclusters as well as the amount of tyrosine phosphorylation (pTyr) or phospho-SLP-76 (pSLP-76) within the microclusters (*Figure 4C–E and F–H*). In contrast, loss of ARPC5L had no effect on the amount of TCR signaling-induced microclusters formed or their pSLP-76 content. In turn, the pTyr intensity in these microclusters was reduced relative to that in control cells, albeit to significantly lower extent than in C5 KO cells. These findings are consistent with the idea that ARPC5 selectively regulates actin polymerization events at the plasma membrane, while ARPC5L is involved in nuclear actin polymerization.

To extend the relevance of ARPC5 isoforms to T cell effector functions, we assessed their contribution to cytokine production in A30.1 cells that produce robust amounts of TNFα and IL-2, 4 or 16 hr post T cell activation. Since we noted that the KO procedure per se impaired cytokine production, ARPC5 isoform gene expression was silenced by shRNA (*Figure 4—figure supplement 3A–C*). Inhibition of Arp2/3 activity by CK869 strongly impaired production of both cytokines (*Figure 4I and J*). Reducing the expression of either ARPC5 or ARPC5L resulted in a significant but partial reduction of TNFα and IL-2 production. This reduction was less pronounced for ARPC5L than ARPC5 in the case of IL-2, possibly reflecting the lower silencing efficiency for ARPC5L (*Figure 4—figure supplement 3B*). Together, these results revealed that TCR proximal signaling in response to CD4 T cell activation is governed by ARPC5-mediated actin polymerization at the plasma membrane that occurs independently from the formation of nuclear actin filaments mediated by ARPC5L. Optimal cytokine production in response to T cell activation, however, requires both isoforms, suggesting a cooperative function between ARPC5 and ARPC5L for proper cytokine gene expression.

## Subcellular localization and association with NFA do not determine the functional specificity of ARPC5 isoforms

We next assessed whether the differential role of ARPC5 and ARPC5L in TCR-induced actin remodeling reflects their distinct cellular distribution. Since immune fluorescence did not allow us to distinguish

between the distribution of endogenous ARPC5 and ARPC5L, we examined the localization of transiently expressed, mCherry-tagged isoforms that functionally rescued our KO cell lines. Both mCherry-tagged ARPC5 and ARPC5L had a diffuse cytoplasmic distribution but were also detected as punctae in the cytoplasm and the nucleus (*Figure 5A*). Staining of endogenous ARPC5 and ARPC5L by the nondiscriminating anti-ARPC5 antibody revealed similar punctae in the cytoplasm and nucleus with an additional localization at the plasma membrane that was particularly pronounced following surface-mediated TCR engagement (*Figure 5—figure supplement 1A and B*). Consistently, immunoblot analysis of nucleo-cytoplasmic fractionations revealed that endogenous ARPC5 and ARPC5L are both present in the nucleus, albeit at lower levels than in the cytoplasm (*Figure 5B*). This distribution of ARPC5 and ARPC5L was unaffected by the loss of expression of the other ARPC5 isoform. To assess the localization of ARPC5.mCherry and ARPC5L.mCherry relative to the NFA network, we applied two-color super-resolution STED microscopy on P/I-stimulated A301 CD4 T cells, which are best suited to visualize endogenous NFA meshwork in T cells (*Tsopoulidis et al., 2019*). Deconvolved and segmented STED images revealed a complex NFA meshwork (*Figure 5C*). ARPC5.mCherry and ARPC5L.mCherry were both detected in discrete spots within the nucleus and ~10% of these spots co-localized with nuclear actin filaments; however, no significant difference was observed between both isoforms (9.9% for ARPC5, 13.2% for ARPC5L, *Figure 5D*). The identity of the ARPC5 isoform involved therefore does not determine the ability of Arp2/3 complexes to associate with actin filaments in the nucleus, but the association of both isoforms with nuclear actin filaments might be required to induce robust cytokine gene expression.

## NFA formation triggered by DNA replication stress involves ARPC5 but not ARPC5L

We next tested whether the selective involvement of ARPC5L is a common principle for Arp2/3-dependent nuclear actin polymerization. In fibroblasts and epithelial cells, Arp2/3 also mediates nuclear actin polymerization in response to DNA replication stress induced by the DNA polymerase inhibitor aphidicolin (APH) (*Lamm et al., 2020*). Consistent with that fact that effective drug concentrations are often elevated in CD4 T cell lines (*Martel et al., 1997*; *Vesela et al., 2017*), we observed efficient formation of NFA in JNLA cells in response to APH starting at 15 μM (*Figure 6—figure supplement 1A*). This response was paralleled by the induction of DNA replication stress as indicated by phosphorylation of the checkpoint kinase CHK-1 (*Figure 6—figure supplement 1B*). NFA formation in response to APH was transient with a maximum of cells (approx. 25%) displaying NFA approximately 90 min post treatment (*Figure 6A–C*, see *Figure 6—figure supplement 1C* for tracks of individual cells). The APH-induced NFA meshwork consists of fewer but thicker F-actin bundles that disassemble more slowly than those induced by T cell activation (*Figure 6B*). Next, ARPC5L and ARPC5 KO cells were stimulated in parallel by T cell activation (P/I or anti-CD3/CD28) or APH. For T cell activation, this confirmed the requirement of ARPC5 for the formation of actin rings (AR) and ARPC5L for the NFA network (*Figure 6—figure supplement 1D*). In contrast, NFA formation in response to APH was indistinguishable to control cells in ARPC5L KO cells but significantly impaired in ARPC5 KO cells (*Figure 6B and D*). NFA induction by T cell activation or DNA replication stress is thus mediated by specific ARPC5 subunit isoforms containing Arp2/3 complexes. Since both isoforms of ARPC5 are present in the nucleus, their nuclear localization alone cannot determine their respective involvement in nuclear actin polymerization events following a specific stimulus. We therefore tested whether this specificity for ARPC5 isoforms is provided by upstream signaling. NFA formation induced by T cell activation is mediated by nuclear calcium transients and can be inhibited by interfering with nuclear calmodulin by expressing the nuclear calmodulin inhibitor calmodulin binding protein 4 (CAMBP4) (*Monaco et al., 2016*; *Tsopoulidis et al., 2019*; *Figure 6E and F* and *Figure 6—figure supplement 1E and F*). In contrast, the nuclear calmodulin inhibitor CAMBP4 did not affect NFA formation or CHK-1 phosphorylation upon APH treatment of JNLA cells (*Figure 6E and F* and *Figure 6—figure supplement 1B*). Similarly, pharmacological inhibitors of downstream effectors of calmodulin including calcineurin inhibitor cyclosporinA (CsA), calmodulin-kinase kinase inhibitor STO609, as well as the calmodulin-kinase II inhibitors KN93 and KN62 did not prevent NFA induction by APH (*Figure 6—figure supplement 1G*). These results suggest that nuclear $Ca^{2+}$-calmodulin acts as a selective trigger for ARPC5L-dependent nuclear actin polymerization upon T cell activation, but is not involved in the formation of NFA upon DNA replication stress.

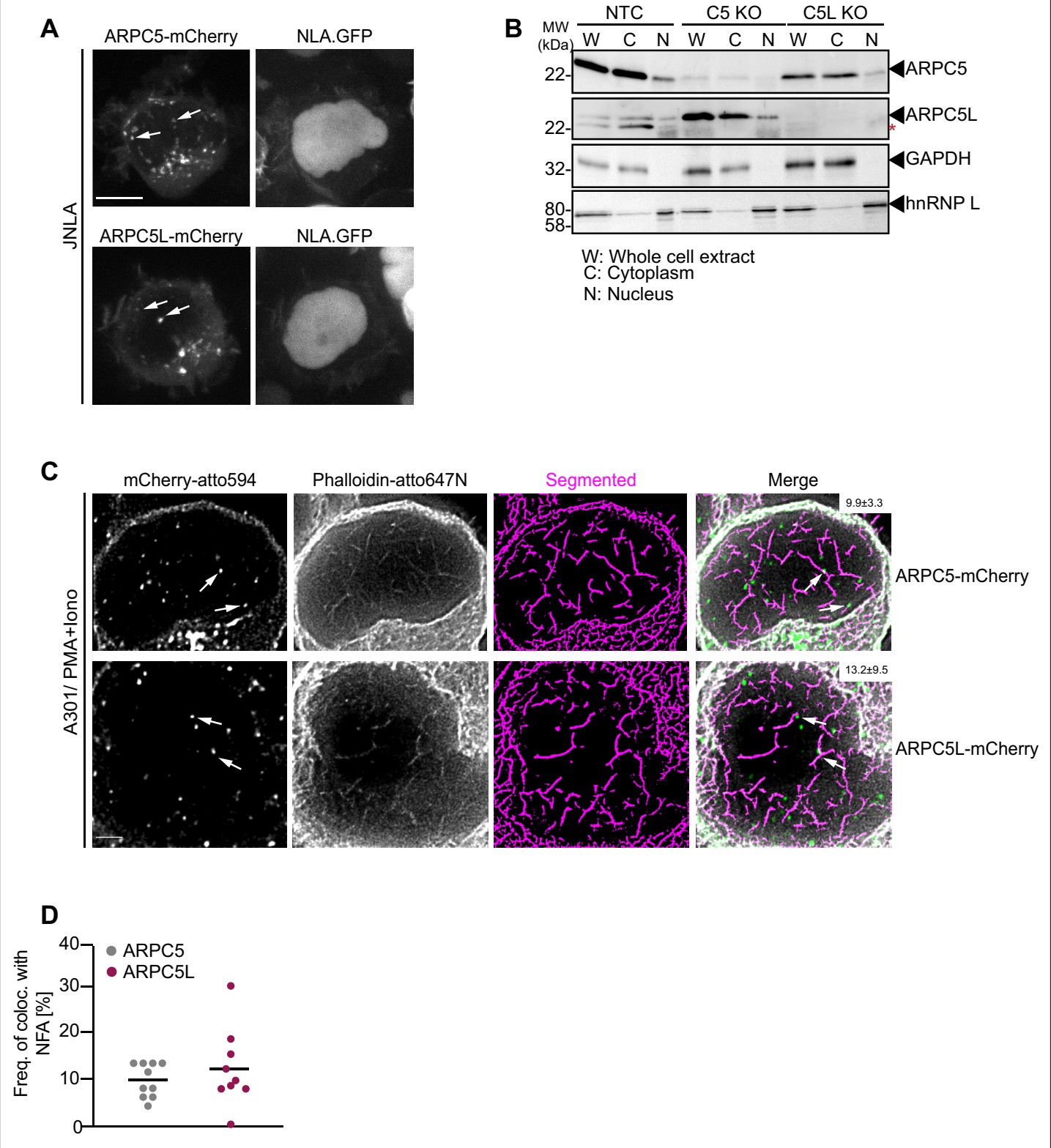

**Figure 5.** Cellular distribution of ARPC5 isoforms. (**A**) Shown are representative spinning-disk confocal images (maximum projection) of ARPC5. mCherry and ARPC5L.mCherry distribution in unstimulated JNLA cells. White arrows point to the respective C5 or C5L punctae seen in the nucleus. Also see related *Figure 5—figure supplement 1A and B*. (**B**) Subcellular distribution of the ARPC5 subunit and its isoform ARPC5L was determined by biochemical fractionation of JNLA cells with respective knockout of either C5 or C5L in the bulk culture. Representative immunoblots reveal levels of ARPC5 isoforms in the whole cell extract (WCE), cytoplasmic (C), and nuclear (N) fractions in the indicated JNLA knockout cells post fractionation.

*Figure 5 continued on next page*

**Figure 5 continued**

GAPDH and hnRNPL were used as markers for cytoplasmic and nuclear compartments, respectively. (**C**) Representative, deconvoluted, and segmented stimulated emission depletion (STED) single-plane images show endogenous nuclear actin filaments (stained with Phalloidin-647N) and ARPC5. mCherry/ARPC5L.mCherry (signal enhanced with anti-mCherry with secondary antibody in atto-594 channel) in A3.01 T cells, stimulated with PMA/ ionomycin (P/I) for 30 s. Arrows (in white) point to the colocalization events. Percent colocalization is mentioned as mean ± SD (in white bar, top right) for each of the isoforms from three independent experiments. Scale bar, 500 nm. (**D**) The dot plot shows the frequency of colocalization of ARPC5 and ARPC5L with nuclear F-actin (from representative STED-deconvolved and segmented super-resolved images shown in *Figure 5C*) in A3.01 cells post 30 s of stimulation with PMA + ionomycin. Each dot represents colocalization events per cell that was analyzed. Source data is avaialble at https://doi. org/10.11588/data/YVYEO8.

The online version of this article includes the following figure supplement(s) for figure 5:

**Figure supplement 1.** Subcellular distribution of ARPC5 isoforms.

## N-WASP selectively drives TCR-induced NFA formation

Since the specificity of distinct nuclear actin polymerization events for distinct ARPC5 isoforms is determined upstream by nuclear $Ca^{2+}$ transients, we assessed whether specific NPFs are involved in this process and focused on class I NPFs with reported roles in the nucleus (*Teitell, 2010*; *Wang et al., 2022*; *Weston et al., 2012*). Unfortunately, we were unable to generate stable WASp KO JNLA cells. However, we managed to obtain bulk KO cultures of N-WASP, WASHC5, an essential subunit of the WASH regulatory complex that is required for the NPF function of WASH (*Jia et al., 2010*), and WAVE2. All anti-NPF antibodies we tested recognized a significant number of unspecific bands, but the comparison of differences between NT and KO cells in signal intensities relative to the GAPDH or Tubulin loading control allowed us to identify NPF-specific NPF protein species, whose expression was significantly reduced in KO cells (*Figure 7A*). Functional characterization revealed the specific involvement of these NPFs in distinct actin polymerization events: while N-WASP was essential for efficient NFA formation in response to P/I, WASHC5 was dispensable for NFA formation and loss of WAVE2 did not affect the frequency of NFA formation but was associated with the formation of shorter nuclear actin filaments (*Figure 7B and C*). A role for N-WASP in NFA formation was also supported by the inhibition by a pharmacological N-WASP inhibitor (*Figure 7—figure supplement 1D and E*). In contrast, N-WASP was dispensable for cell spreading and AR formation induced by TCR signaling while loss of WASHC5 or WAVE2 significantly impaired these processes (*Figure 7D and E* and *Figure 7—figure supplement 1F*). Interestingly, NFA formation in response to APH was unaffected in N-WASP, WASHC5, or WAVE2 KO cells (*Figure 7F and G*) and is hence governed by other upstream regulators. NFA formation by TCR signaling is thus governed by a selective pathway that depends on ARPC5L containing Arp2/3 complexes that are activated by N-WASP and nuclear $Ca^{2+}$ transients. Interestingly, while expression levels of Arp2/3 subunits and isoforms were overall unaltered in NPF KO cells, ARPC5L levels were significantly reduced in N-WASP KO cells, suggesting a co-dependency for stable complex formation (*Figure 7—figure supplement 1A–C*). Functional coupling of ARPC5L and N-WASP may thus involve a regulatory mechanism at the level of protein expression/stability.

## Discussion

A main achievement of our study is that we were able to capture nuclear and cytoplasmic actin polymerization triggered by a common stimulus (T cell activation) or nuclear actin polymerization by different stimuli (T cell activation, DNA replication stress) within the same CD4 T cell system. This establishes CD4 T cells as an ideal cell type for studying the complex regulation of actin polymerization in different cellular compartments and in response to various stimuli. Using this system, we identified that the response to various stimuli is mediated by distinct Arp2/3 complexes containing ARPC5L or ARPC5. Given that (i) nuclear actin polymerization events, such as those induced by TCR signaling or APH-mediated induction of DNA replication stress, are mediated by distinct Arp2/3 complexes with preferences for specific ARPC5 subunits and (ii) Arp2/3 complexes containing both isoforms are functional in the nucleus, it appears that the specificity for a particular ARPC5 isoform is not determined by its subcellular distribution. Rather, the nature of the stimulus is critical for the selective induction of actin polymerization by ARPC5 or ARPC5L containing Arp2/3 complexes (see schematic model in *Figure 8*). Our results define responsiveness to nuclear calcium-calmodulin signaling and

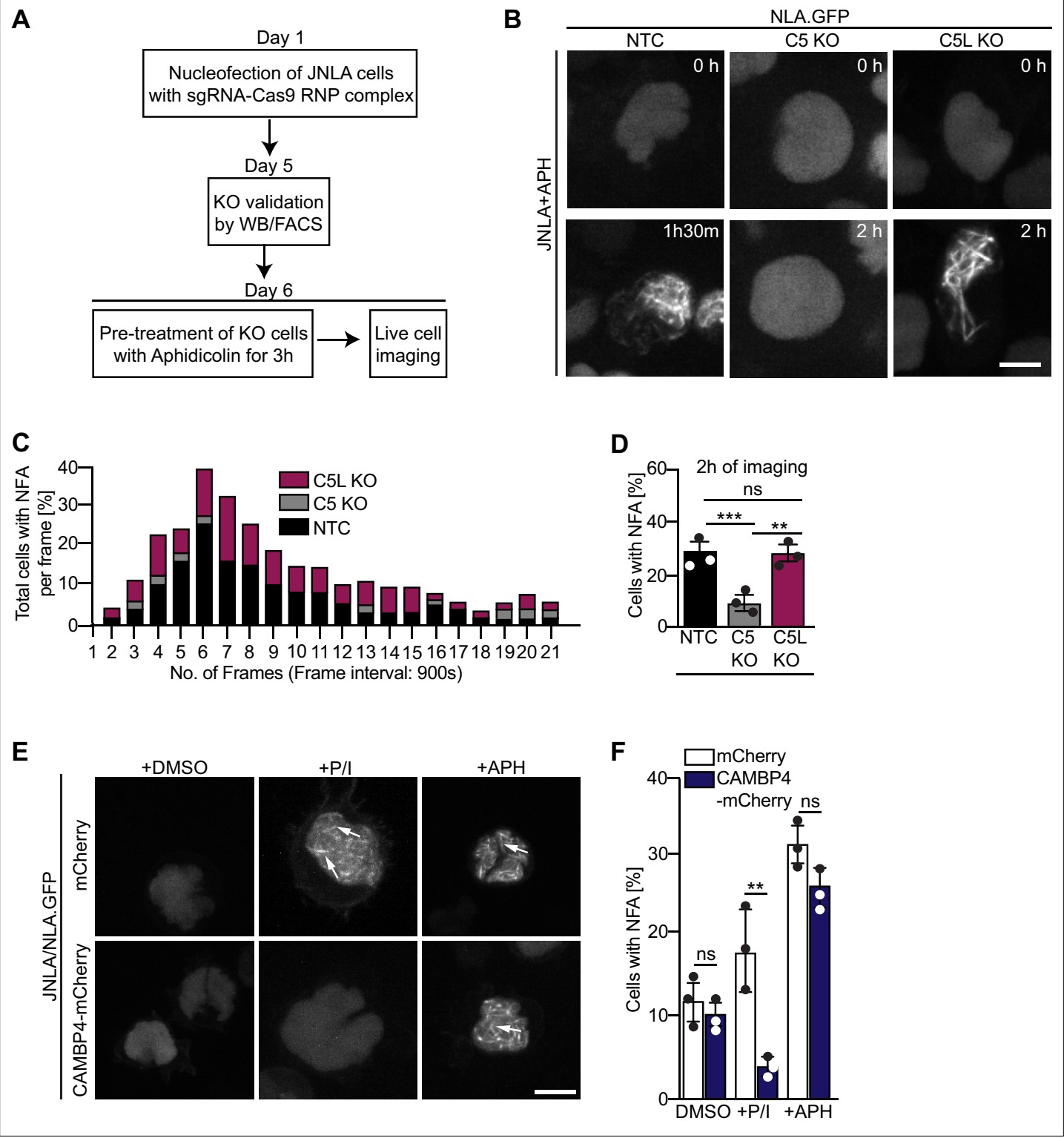

**Figure 6.** Differential role of ARPC5 isoforms in replication stress-mediated nuclear F-actin (NFA) formation. (**A**) Schematic of experimental setup showing timeline of knockout (KO) generation and induction of replication stress in the JNLA KO cells using aphidicolin (APH). See related *Figure 6—figure supplement 1A*. (**B**) Shown are representative spinning disk confocal still images (maximum projection) of the APH pretreated KO or control cells. The movies were acquired for 5 hr with acquisition every 15 min post pretreatment of cells with APH. The stills at the indicated time points are representative of the time point where the maximal NFA burst has been observed in each condition. (**C**) Stacked bar graph (denoted by three different colors for each condition) shows the kinetics of NFA burst throughout 5 hr of live-cell imaging duration, with maximum NFA burst observed within

*Figure 6 continued on next page*

*Figure 6 continued*

the first 2 hr. (**D**) Quantification of the % of cells with NFA bursts (within the first 2 hr of imaging) post replication stress induction in control and KO cells. Each data point indicates the mean value of an independent experiments with at 40–60 analyzed per condition with indicated mean ± SD from three independent experiments. Statistical significance was calculated using one-way ANOVA (Kruskal–Wallis test). Also see related *Figure 6—figure supplement 1C* for single-cell tracks. (**E**) JNLA cells transduced with either mCherry (control) or CAMBP4.NLS-mCherry were pretreated with solvent control (DMSO) and were either activated by PMA + ionomycin (P/I) or treated with APH for induction of replication stress for 3 hr prior to live-cell imaging. Shown are maximum projection of representative spinning-disk confocal still images (showing the time frame where maximum NFA burst was observed) in the DMSO control compared with either P/I or APH-mediated NFA bursts (white arrows) in the presence and absence of nuclear calmodulin. Movies for visualizing replication stress were acquired for 5 hr with acquisition every 15 min post pretreatment of cells. Whereas movies for visualizing P/I activation-induced NFA were acquired for 5 min with acquisition every 15–30 s. Also see related *Figure 6—figure supplement 1E and F*. (**F**) Quantification of NFA in the abovementioned conditions was performed; each data point indicates the mean value of an independent experiment with at least 30 cells analyzed per condition with indicated mean ± SD from three independent experiments. Statistical significance was calculated using Welch's *t*-test. *p≤0.0332, **p≤0.0021, ***p≤0.0002, and ns: not significant. Scale bar, 3 µm. Also see related *Figure 6—figure supplement 1B*.

The online version of this article includes the following figure supplement(s) for figure 6:

**Figure supplement 1.** Nuclear F-actin (NFA) formation induced by aphidicolin (APH) is not dependent on calcium signaling in CD4 T cells.

regulation by the NPF N-WASP as specific triggers of ARPC5L containing complexes. The selective involvement of N-WASP likely reflects that its ability to activate Arp2/3 can be achieved by calcium-calmodulin. This may occur directly via the interaction of N-Wasp with calmodulin, which is known to activate the NPF. Alternatively, the Ras GTPase-activating-like protein IQGAP has been reported to be activated by binding to calmodulin. Activated IQGAP could in turn induce N-Wasp by direct interaction or via activation of Rho-GTPases upstream of N-Wasp (*Le Clainche et al., 2007*; *Miki et al., 1996*; *Pelikan-Conchaudron et al., 2011*; *Rhoads and Friedberg, 1997*). It will be important to dissect which of these scenarios allow ARPC5L to determine the sensitivity of Arp2/3 to regulation by N-WASP-calcium-calmodulin at the molecular level. The identity of the ARPC5 isoform could potentially influence the affinity of Arp2/3 to N-WASP independently of its activation by calcium-calmodulin. However, existing structural and biochemical data cannot explain a direct involvement of ARPC5/C5L in interactions with nucleation-promoting factors (NPFs) (*von Loeffelholz et al., 2020*). ARPC5 isoforms thus likely undergo specific interactions with additional interaction partners that govern the susceptibility of Arp2/3 complex to activation via N-WASP-calcium-calmodulin. Indeed, a recent study by Fäßler and colleagues reports that in migrating fibroblasts the identity of the involved ARPC5 isoform affects the positioning of the Arp2/3 effectors Mena/VASP and thereby affects filament polymerization velocity (*Faessler et al., 2022*). In addition to defining this mechanism, it will be interesting to determine whether induction of ARPC5L-containing complexes by calcium-calmodulin can also occur in the cytoplasm and how the observed stabilization of ARPC5L expression by N-WASP contributes to the regulation of this pathway. Our attempts to gain spatio-temporal information on Arp2/3 complexes containing specific ARPC5 isoforms with respect to these NPFs were hampered by high background staining of anti-NPF antibodies and toxicity of co-overexpression of NPF and C5 isoform and new experimental approaches need to be developed to tackle this issue.

In contrast, Arp2/3 complexes containing ARPC5 such as those involved in DNA replication stress trigger nuclear actin polymerization independently of calcium-calmodulin-N-WASP and are likely regulated by another NPF. It is tempting to speculate that the regulation of ARPC5L containing Arp2/3 complexes by nuclear calcium-calmodulin reflects the requirement for rapid conversion of an extracellular signal, for example, to elicit a transcriptional response. In line with this scenario, calcium-mediated induction of nuclear actin assembly by the formin INF2 in mouse fibroblasts also represents a rapid response to conversion of an extracellular signal (*Wang et al., 2019*). In contrast, DNA replication stress provides a signal from within the nucleus without the need for a fast second messenger. Notably, T cell activation or DNA replication stress induces NFA networks of different filament morphology and dynamics. These architectural differences may reflect physiological properties of Arp2/3 complexes with different ARPC5 isoforms and translate into distinct functional roles. In this scenario, thin/dynamic filaments may mediate transcriptional regulation following TCR engagement while thicker and more stable filaments could exert mechanical functions during DNA repair. The ARPC5 isoform subunit specificity of Arp2/3 activity in the cytoplasm and the nucleus identified herein opens avenues for dissecting the roles of the transient nuclear actin filament network such as the control of gene expression triggered by T cell activation. The currently available data link

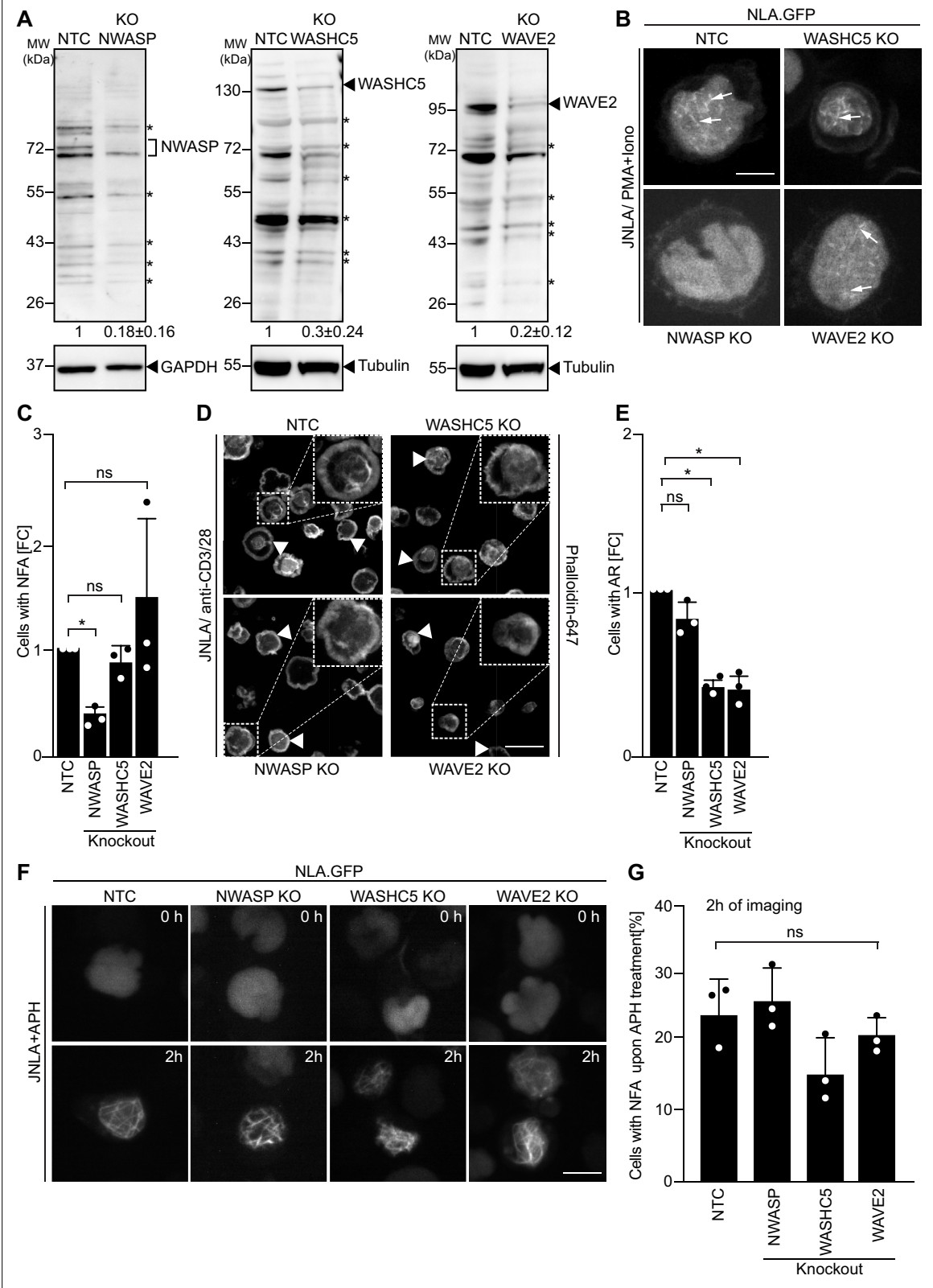

**Figure 7.** Differential involvement of class I nucleation-promoting factors (NPFs) in T cell activation and replication stress-mediated nuclear F-actin (NFA) formation. (**A**) Representative immunoblots show knockout (KO) of NWASP, WASHC5, and WAVE2 class I NPFs, respectively, in JNLA cells. Identical amounts of cell lysates were loaded per lane and signal intensity judged based on the loading control (GAPDH or Tubulin). Black arrowheads or asterisks mark specific and unspecific bands, respectively, as identified based on signal reduction in the KO cell line. The numbers indicated below

*Figure 7 continued*

the respective blots represent the mean ± SD values from three independent experiments, based on the densitometric quantification of the bands, normalized to the loading control and compared to the non-targeting control (NTC) protein levels (which is set to 1). (**B**) Representative spinning-disk confocal still images (maximum projection) of JNLA cells with indicated NPF KO showing NFA formation post activation with PMA/ionomycin (P/I). Movies were acquired for 5 min post PI addition with acquisition every 30 s. Arrows point to the NFA. Scale bar, 3 µm. (**C**) Quantification of NFA formation in respective NPF KO cells relative to the NTC-treated cells. Each data point indicates the mean value of an independent experiments with 30 analyzed per condition with the indicated mean ± SD from three independent experiments. (**D**) Representative confocal images indicate averaged intensity projections of Phalloidin-647-stained F-actin ring (AR) formation in fixed/permeabilized JNLA cells with respective NPF KO upon activation on coverslips coated with anti-CD3+CD28 antibodies. Arrowheads point to the AR at the PM upon TCR activation, and the dotted box in white shows the cell in zoomed view in the inset (top right). Scale bar, 5 µm. (**E**) Quantification of Phalloidin-stained AR formation in KO cells relative to the NTC-treated cells. Each data point indicates the mean value of an independent experiment with at least 100 cells analyzed per condition with indicated mean ± SD from three independent experiments. (**F**) Shown are representative spinning-disk confocal still images (maximum projection) of the aphidicolin (APH)-treated NPF KO or control cells, respectively. The movies were acquired for 5 hr with acquisition every 15 min post pretreatment of cells with APH. The stills at the indicated time points are representative of the time point where the maximal NFA burst has been observed in each condition. (**G**) Quantification of the % of cells with NFA bursts (within first 2 hr of imaging) post replication stress induction in control and KO cells. Each data point indicates the mean value of an independent experiment with 40–60 cells analyzed per condition with indicated mean ± SD from three independent experiments. One-way ANOVA with Kruskal–Wallis test was used to determine statistical significances, where $*p \leq 0.0332$ and ns: not significant. Also see related *Figure 7—figure supplement 1A-F*. Source data is avaialble at https://doi.org/10.11588/data/YVYEO8.

The online version of this article includes the following figure supplement(s) for figure 7:

**Figure supplement 1.** Molecular characterization of Arp2/3 complex upon nucleation-promoting factor (NPF) knockout (KO) in JNLA cells.

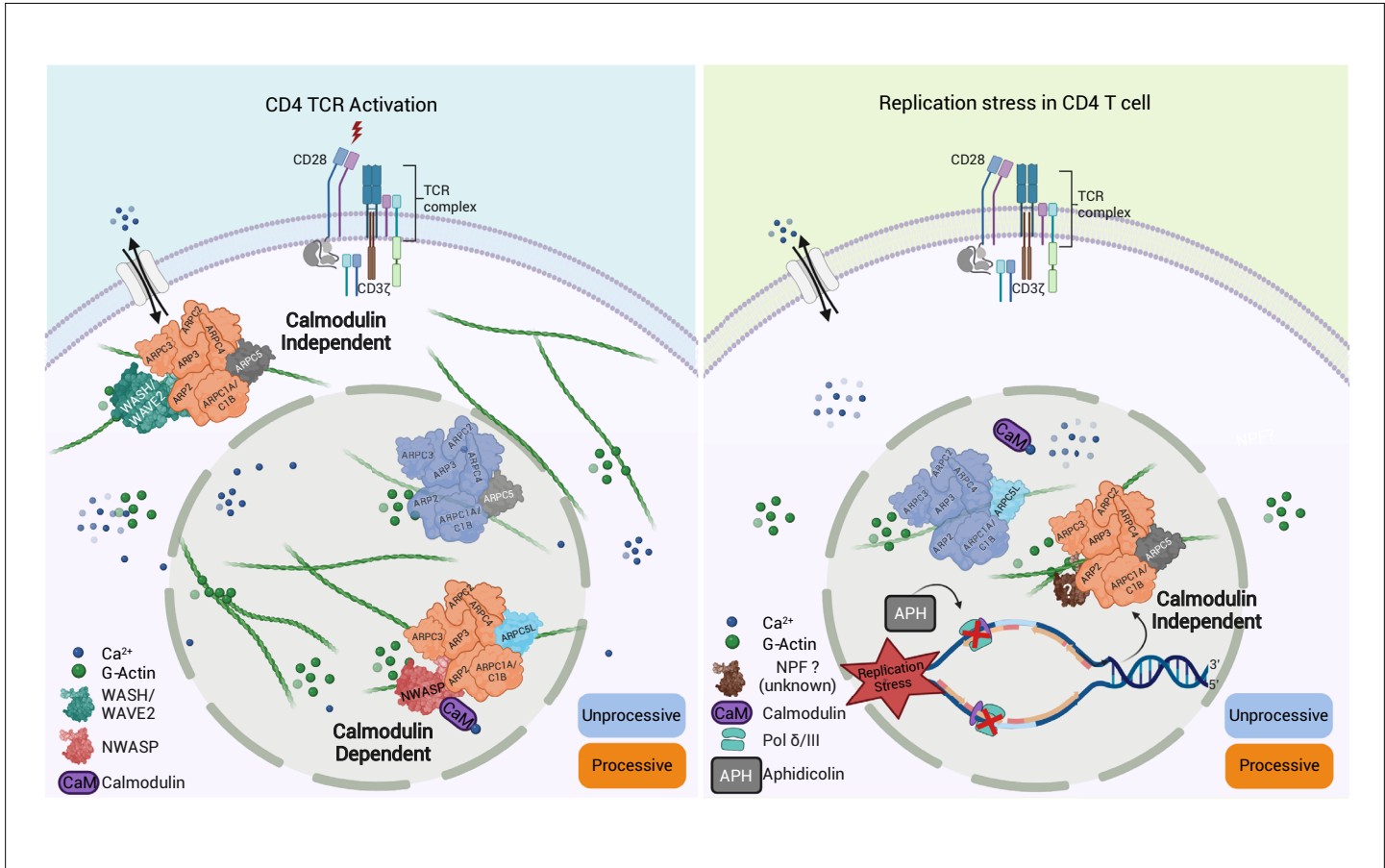

**Figure 8.** Graphical summary of our findings. Schematic model for Arp2/3-dependent differential regulation of actin dynamics induced upon TCR engagement (left) compared to the induction of replication stress by aphidicolin (APH) (right).

the visualization of nuclear actin dynamics in individual cells to the analysis of cellular functions in bulk populations. Our single-cell expression analysis revealed a good correlation between ARPC5L expression and NFA formation. However, the frequency of cells that respond to T cell activation with cytokine production is significantly lower than that of cells displaying NFA formation, suggesting that NFA formation is necessary but not sufficient for cytokine production and might also regulate other T cell effector programs. Dissecting the specific role of NFA in gene expression control in future studies will require analyzing the transcriptional activity of single cells of which the NFA response has been recorded previously.

The preference for distinct ARPC5 isoforms thus likely adjusts the activity of Arp2/3 complex to divergent actin polymerization events that are triggered by specific upstream signals. Similarly, Arp2/3 complexes containing different ARP3 isoforms were recently shown to be differentially regulated (*Abella et al., 2016*; *Galloni et al., 2021*). The subunit isoform composition of Arp2/3 complexes thus emerges as an important parameter that allows Arp2/3 to mediate distinct actin polymerization events tailored to specific activation signals at selected subcellular sites.

## Materials and methods
### Cells and reagents

All cell lines were received by ATCC, were tested negative for mycoplasma, and were recently authenticated via STR profiling. HEK 293T cells were cultured in DMEM high glucose plus 10% feline bovine serum (FBS, Millipore), 100 U/ml penicillin, and 100 µg/ml streptomycin. Primary T cells, Jurkat Tag cells (JTAgs), and CLEM-derived A3.01 cells were cultured in RPMI containing 10% FBS and 1% penicillin-streptomycin and GlutaMAX-I (Gibco). All experiments performed in JTag cells stably expressing nuclear lifeact-GFP (JNLA) were obtained as described previously in *Tsopoulidis et al., 2019*. All cell lines were cultivated according to their ATCC (https://www.atcc.org) guidelines. For visualization of nuclear F-actin, A3.01 or JNLA were washed thoroughly with PBS, adjusted to a cell density of 3E5/ml, and incubated overnight in RPMI (phenol-Red free medium, Gibco) containing 0.5% (A3.01) or 10% (JNLA) FBS. For immunofluorescence (IF) microscopy: F-actin was stained with Phalloidin Alexa Fluor 488 or atto-AF488 (Thermo Fisher). Alexa antibodies for IF such as goat anti-mouse Alexa Fluor 568, goat anti-rabbit Alexa Fluor 647, and goat anti-rabbit Alexa Fluor 568 were obtained from Thermo Fisher Scientific. The following anti-CD3 (clone HIT3a against CD3ε; BD Pharmingen) and mouse anti-CD28 (CD28.2, BD Pharmingen) were used at 1:100 dilution for coating coverslips/GBDs to make stimulatory surface for T cell activation. Antibodies used for immunoblotting were mouse-anti-ARP3, 1:10,000 (cloneFMS338, Sigma), mouse anti-Arp2, 1:1000 (sc-137250, SCBT), mouse anti-p16-ARC, 1:500 (#305011, Synaptic Systems), rabbit anti-ARPC5L, 1:1000 (GTX120725, GeneTex), rabbit anti-ARPC1A, 1:500 (#HPA004334, Sigma), mouse anti-ARPC1B, 1:500 (SCBT), rabbit anti-ARPC2, 1:1000 (EPR8533, Abcam), mouse anti-ARPC3, 1:500 (#HPA006550, Sigma-Aldrich), mouse anti-ARPC4, 1:500 (#NBP1-69003, Novus Biologicals), mouse anti-WAVE2, 1:500 (#sc-373889, SCBT), rabbit anti-NWASP/WASL, 1:1000 (#HPA005750, Sigma-Aldrich), rabbit anti-WASHC5, 1:250 (#HPA070916, Sigma-Aldrich), mouse anti-Tubulin, 1:1000 (#373, DM1A, CST), rabbit anti-GAPDH, 1:2500 (#2118, 14C10, CST), mouse anti-GAPDH, 1:2000 (#MCA4740, Bio-Rad), mouse anti-hnRNPL, 1:2000 (ab6106, Abcam), mouse anti-mCherry, 1:1000 for WB and 1:500 for IF (NBP1-96752), rabbit anti-mCherry, 1:1000 for WB and 1:500 for IF (ab167453), rabbit anti-pTyr, 1:100 (#sc18182, SCBT), and rabbit anti-pSLP76, 1:1000 (#ab75829, Abcam). HRP-coupled secondary rabbit or mouse antibodies for immunoblotting were obtained from Jackson ImmunoResearch and were used at a dilution of 1:5000 for all samples. The secondary Alexa fluorescent coupled antibodies (either mouse or rabbit) used for IF staining were obtained from Invitrogen and were used at a dilution of 1:1000.

For live-cell imaging and STED microscopy: glass-bottom dishes (GBD) with 35 mm plate diameter, 14 mm glass diameter, thickness 1.5 (Mattek Corporation), and µ-slide 8-well glass bottom chambers (Ibidi) were used along with poly-lysine (Sigma), coated at a concentration of 0.01% in sterile-filtered water. Phalloidin atto-647N used for STED imaging was bought from ATTO-TEC GmbH (AD 647N-81) and used at a dilution of 1:500 in blocking buffer containing 3% FCS/PBS for immunofluorescence staining, whereas for super-resolved STED imaging Phalloidin was dissolved in 5% FCS/cytoskeleton buffer (1×).

## Preparation of primary CD4 T cells

For the isolation of primary human CD4 T cells, human Buffy Coats from anonymous healthy donors were obtained from the Heidelberg University Hospital Blood Bank. CD4+ T cells were isolated by negative selection with the RosetteSepTM Human CD4+ T Cell Enrichment Cocktail and separated by Ficoll gradient centrifugation, resulting in homogenous populations of CD4+ T cells with a purity of 90–95% as assured by flow cytometry. Cells labeled as 'Resting' were cultured for 72 hr in complete RPMI media containing recombinant human IL2 (Biomol #155400.10) at 10 ng/ml final concentration. Whereas the cells labeled as 'Activated' were cultured for 72 hr in complete RPMI media containing recombinant human IL2 (Biomol #155400.10) at 10 ng/ml final concentration along with Dynabeads at a ratio of 25 µl human anti-CD3/28-labeled Dynabeads/10 million cells (#11132D, Gibco).

## Agonists and inhibitors

The following chemicals were used at the indicated concentrations: ionomycin (Iono, 2 µM), phorbol 12-myristate 13-acetate (PMA, 162 nM), CK-869 (100 µM), KN-93 (0.25 µM), KN-62 (2.5 µM), STO-609 (5 µM), and aphidicolin (15 µM) were all obtained from Sigma-Aldrich, whereas cyclosporine A (1 µM) and 187-1 (NWASPi, 3 µM) were obtained from TOCRIS Bioscience. Of note, aphidicolin and NWASPi are very unstable and lose activity within 7 d and 2 d, respectively, post reconstitution and storage at –20°.

## Expression plasmids

pLVX vector expressing either human ARPC5/C5L cDNA fused to a mCherry fluorescent reporter or just the mCherry alone were a kind gift from the lab of M. Way, generated as described in *Abella et al., 2016* with puromycin (1.5 µg/ml) used as a selection marker. Plasmids expressing mCherry alone or mCherry conjugated to CAMBP4 in the pWPI backbone were used as described (*Tsopoulidis et al., 2019*) and were selected using blasticidin (5 µg/ml) for 48–72 hr. For the stable expression of shRNAs, gene-specific target sequences (mentioned in the MDAR report) were obtained from the publicly available TRC cloning portal of Sigma. These sequences were cloned into the lentiviral vector pLKO.1-puro (Addgene) as described in *Tsopoulidis et al., 2019* and were selected with puromycin (1.5 µg/ml) for 48 hr post transduction.

## Live-cell imaging of actin dynamics

Live imaging of actin dynamics was performed with a Nikon Ti PerkinElmer UltraVIEW VoX spinning disc confocal microscope equipped with a perfect focus system (PFS), a ×60 oil objective (numerical aperture, 1.49), Hamamatsu ORCA-flash 4.0 scientific complementary metal-oxide semiconductor camera, and an environmental control chamber (37°C, 5% $CO_2$), as described earlier in *Tsopoulidis et al., 2019*. Acquisition settings varied depending on the total acquisition time required for each experimental question/setup. The following are the acquisition settings for short-term imaging: exposure time, 300 ms; frame rate, 6–10 frames/s, number of Z planes, 10; Z-stack spacing, 0.5 µm; 488 nm, laser power between 4.5 and 5.5%; and total acquisition time, 3–10 min. Jurkat cells stably expressing nuclear lifeact.GFP (JNLA) were always washed with PBS and split 24 hr before the experiment to a density of $3 \times 10^5$/ml. The next day $3 \times 10^5$ cells were harvested, washed with PBS, and resuspended in 100 µl reconstituted RPMI containing 10% FCS. For imaging the actin dynamics of JNLA cells falling on stimulatory surface (coated with anti-CD3+CD28 antibodies), the PFS system was adjusted first with a low amount of highly diluted cells placed on the coated glass-bottom dish. A single cell was centered to the field of view, and the PFS was adjusted to automatically focus on the glass-cell contact site. Subsequently, the stage was moved to a cell-free area, and 100 µl of the cell suspension ($3 \times 10^5$/100 µl) was added to the glass-bottom dish with simultaneously recording cells while making contact with the glass surface.

For imaging of actin dynamics in cells resting on the glass surface, 100 µl of cells ($3 \times 10^5$ /ml) was plated on polyK-coated glass-bottom dishes, allowed to adhere for 5 min, and then stimulated with P/Iin RPMI media.

## Super-resolution imaging of nuclear actin

A3.01 T lymphoblastoid cells were washed and adjusted to cell density of 0.35 million cells/ml a day prior to the experiment, as described above. The next day 0.6 million cells were collected/well of an

8-well chambered dish, washed once with PBS, and resuspended in of RPMI (phenol-free) containing 0.5% FBS. Cells were allowed to adhere for 5 min on polyK-coated 8-well chamber glass-bottom dish. Stimulation was performed by adding 100 µl of PMA/iono solution dropwise to the cell suspension. Cells were activated for 30 s and then permeabilized and stained with 100 µl permeabilization solution containing 0.3% Triton X-100 + Phalloidin-Alexa Fluor 488 (1:2000) in fresh 1× cytoskeleton buffer (10 mM MES, 138 mM KCl, 3 mM MgCl, 2 mM EGTA, and 0.32 M sucrose [pH 6.1], made in-house) for 30 s. Addition of low amounts of phalloidin at this step is required to stabilize nuclear actin filaments during permeabilization/fixation. Cells were fixed with 1 ml of 4% methanol-free formaldehyde (Pierce) in 1× cytoskeleton buffer and incubated for 25 min at room temperature (RT) in the dark. Subsequently, the fixed cells were washed twice with cytoskeleton buffer, blocked with 5% bovine serum albumin (BSA) prepared in 1× cytoskeleton buffer, and stained with 1:500 Phalloidin atto-647N in 1× cytoskeleton buffer for 1 hr at RT or overnight at 4°C. Additionally, to enhance the mCherry signal of the C5/C5L-mCherry expression constructs in the cells, primary antibody staining using anti-mCherry antibody (1:500) was performed overnight in blocking buffer as mentioned above. This was followed by multiple washing steps in 1× cytoskeleton buffer and staining with secondary antibody conjugated with an atto-568 dye, for 1 hr at RT. Phalloidin atto-647N was added at this step with the secondary antibody to stain the endogenous nuclear actin filaments, which were observed in ~40–60% of the cells.

STED microscopy was performed on an Expert Line STED system (Abberior Instruments GmbH, Göttingen, Germany) equipped with an SLM-based easy3D module and an Olympus IX83 microscope body using a ×100 oil immersion objective (NA, 1.4; Olympus UPlanSApo). STED images were acquired using the 590 nm (ARPC5/ARPC5L signals) and 640 nm (actin filament signals) excitation laser lines in the line sequential mode with corresponding 615/20 and 685/70 emission filters placed in front of avalanche photodiodes for detection. 775 nm STED laser (15% of the maximal power of 3 mW) was used for depletion with pixel dwell time of 10–15 µs, 15 nm xy sampling and 9× accumulation. To increase the signal-to-noise and facilitate subsequent image segmentation and quantification, STED images were restored with Huygens Deconvolution (Scientific Volume Imaging) using Classic Maximum Likelihood Estimation (CMLE) algorithm and Deconvolution Express mode with 'Conservative' settings.

To segment actin filaments and ARPC5/ARPC5L signals in obtained STED images, we trained a Random Forest classifier using ilastik (*Berg et al., 2019*) autocontext workflow that predicts semantic class attribution (signal or background) for every pixel. The training set of data was arbitrary selected and very sparsely labeled (<0.1% of total pixels were manually categorized into 'signal' and 'background' categories). Obtained machine learning algorithm was used applied to all acquired images ensuring an unbiased signal segmentation across all experiments. This allowed the quantification of the number of ARPC5/ARPC5L signals colocalizing with nuclear actin filaments by visual inspection of binary (segmented) images.

## Imaging actin dynamics at the Immune synapse post CK-inhibitor treatment

To distinguish B cells from T cells before mixing them together for live imaging, Raji B cells were stained with Cell trace Deep Red (10 µM, Thermo Fisher) at 1:1000 dilution for 1 hr and simultaneously loaded with SEE (Toxin Technology) at a concentration of 5 µg/ml, in RPMI complete media for 30 min at 37°C and subsequently washed and resuspended in 10% FBS containing RPMI at a concentration of $5 \times 10^4$ cells in 100 µl. JNLA cells were washed and adjusted a day before as described above. 24 hr later $1 \times 10^6$ were harvested, washed in PBS, and resuspended in 100 µl RPMI complete media containing either DMSO or the CK869 for 1 hr at 37°C. The media are replenished after 1 hr with fresh media containing either the solvent or the inhibitor such that the cells are at a final density of $5 \times 10^4$ cells in 100 µl. 100 µl of the treated JNLA cells are plated on a poly-lysine-coated GBDs. Approximately 5–10 regions on the GBDs were selected for live-cell imaging using the spinning-disk confocal microscopy as described above. Imaging was started and 100 µl Raji B cells were added dropwise onto the T cell suspension while the image acquisition was ongoing. The following are the acquisition settings for the imaging: exposure time, 200–300 ms; frame rate, 6–10 frames/s, number of Z planes, 3; Z-stack spacing: 1–1.5 µm; 488 nm, laser power 5.5%; and total acquisition time, 30 min with acquisition every 30 s/XY position.

## Flow cytometric staining and measurements

For determining the changes in levels of surface CD3 expression, 6 million Jurkat cells were divided into three groups as follows: unstimulated cells and cells pretreated with either solvent (DMSO) or CK869 for 30 min. This was followed by stimulating DMSO and CK869 pretreated cells with PMA + ionomycin for either 5 min or 30 min. Upon respective time points of stimulation, the cells were stained for anti-human CD3-FITC antibody (1:100) in MACS buffer (1×) for 15 min on ice followed by a washing step in MACS buffer (1×) before fixing the cells for 15 min in BD Cytofix buffer (100 µl/million cells) at RT. All conditions were washed twice in MACS buffer (1×) following fixation and resuspended in 300 µl cold-PBS (1×) for flow cytometric measurements on BD FACS Celesta immediately or the next day.

For intracellular cytokine measurements, cells from A.301 lymphoblastoid cell line were transduced with lentiviruses to express the shRNA against the scrambled control and the ARPC5 isoforms C5 or C5L. The protocol for transduction and antibiotic selection followed was similar to the method mentioned earlier. 72 hr post transduction and 24 hr post puromycin selection (in 1.5 µg/ml), A.301 cells were washed extensively in tissue culture grade PBS (1×) and adjusted to a cell density of 1 × 10⁶ cells/ml overnight. The next day 2 × 10⁶ cells were either left untreated, pretreated with for 30 min with DMSO or CK869 (for control and parental cells), and pretreated with DMSO (30 min) for cells expressing the shRNA against ARPC5 isoforms. PMA + ionomycin (PI) stimulation was added to the pretreated cells in the presence or absence of DMSO or CK869 for 4 hr in the presence of monensin (1:1000 in RPMI) to detect TNF$a$. Similarly, for detection of IL-2, PI stimulation of the pretreated cells in the presence or absence of DMSO or CK869 was performed for 16 hr, with addition of monensin (1:1000 in RPMI) done around 12 h post stimulation.

Cells were fixed in 100 µl of BD Cytofix buffer for 15 min at RT upon respective periods of PI stimulation, followed by a washing step in 1 ml of BD Perm/Wash buffer (1×) and 10 min of permeabilization in 100 µl of BD Perm Buffer III. Antibody staining with antibodies against anti-human TNF$a$-BV421, anti-human IL2-APC, or antibodies against respective isotype as controls was performed in 100 µl (1:100 antibody dilution) of 1× BD Perm/Wash solution at 4° for 30 min followed by a washing step and resuspension in 200 µl of PBS (1×). Intracellular cytokine measurements with BD Celesta were proceeded with immediately, after staining on the same day.

## RNA extraction and quantitative PCR (qPCR)

For RNA extraction NucleoSpin RNA II kit (Macherey-Nagel) was used. 10 × 10⁶ cells were collected per condition/per cell line, washed with cold PBS once, and their pellets were stored at –80°C for maximum 2–3 wk. RNA extraction was done following the manufacturer's protocol. After RNA quantification by UV/VIS spectrometry (Nanodrop), between 500 ng and 1000 ng of total RNA was reverse transcribed using the SuperScriptII (Invitrogen) according to the manufacturer's instructions. 1:10 dilution of the cDNA in RNAse-free water was used for qPCR reaction using the SYBR green PCR master mix (Life Technologies), and reactions were performed on a Quant Studio1 sequence detection system (Applied Biosystems) using the following program: 50°C for 2 min, 95°C for 10 min, and 40 cycles of 95°C for 15 s and 60°C for 1 min. GAPDH (glyceraldehyde-3-phosphate dehydrogenase) mRNA was used for normalization of input RNA wherever needed or mentioned. The primer sequences used are mentioned in the MDAR report and were obtained from the publicly available online PRIMER BANK database.

## Immunoblot analysis

1 × E6 cells/condition were collected and lysed in lysis buffer (50 mM Tris-HCl [pH 7.4], 75 mM NaCl, 1 mM EDTA, 1 mM NaF, and 0.5% NP-40) with a freshly added protease inhibitor cocktail and sodium vanadate and subjected to nine cycles (30 s ON–10 s OFF) of ultrasonication (Bioruptor Plus; Diagenode). The sonicated samples are then spun down, and the supernatant is collected for protein estimation using the microBCA kit (Pierce). 10 µg of protein is then mixed with 1× sample buffer (10% sucrose, 0.1% bromophenol blue, 5 mM EDTA [pH 8.0], 200 mM Tris [pH 8.8]), and boiled at 95°C for 10 min. The lysates are then run on either self-made 10–15% SDS-PAGE gel or on pre-casted

Invitrogen NuPAGE 4 bis 12 %, Bis-Tris, 1.0–1.5 mm, Mini-Protein-Gel, followed by blotting with Transblot PVDF membranes (Bio-Rad) for 15 min, blocked in 5% BSA in TBS-T for 1 hr before probing with the primary antibodies overnight at 4°C. Secondary antibodies conjugated to HRP were probed for 1–1.5 hr at RT the next day following 3× intensive TBST washing of the unbound primary antibodies. Enhanced chemiluminescence (ECL)-based detection using the WesternBright Sirius Chemiluminescent Detection Kit (Advansta) was performed. Densitometric quantification was performed manually using Fiji (gel analysis tool).

## Lentivirus production

For small-scale production of lentiviral vectors containing shRNA constructs or the pLVX-expression plasmids, $3 \times 10^5$ HEK 293T cells were seeded per 6 cm dish (2 ml media per well) 24 hr before transfection. Transfection was performed using JetPEI (VWR International) with 1.5 µg of Vector DNA, 1 µg of psPAX2, and 0.5 µg of vesicular stomatitis virus G protein plasmid (pMD2.G) and 0.2 µg pAdvantage per well of a 6-well. Virus supernatants were harvested after 48 hr, filtered through 0.45-µm-pore-size filters (Roth), and used immediately for transduction.

For the generation of stable T cell lines expressing the C5/C5L-mCherry constructs or primary human T cells expressing the Lifeact-GFP constructs, five 15 cm Petri dishes were prepared with $2.5 \times 10^6$ HEK293T cells/dish in 22.5 ml medium. The transduction solution was prepared in a 50 ml reaction tube, containing 112.5 µg vector, 40 µg (pMD2.G), 73 µg psPAX2, 25 ml NaCl, and 500 µl JetPEI. The transduction solution was mixed and incubated at RT for 20 min. For every dish, 5 ml of the solution was used. The dishes were incubated for 4 hr at 37°C before changing the media. The supernatant containing virus particles was collected after 48 hr and filtered via 0.45 µm filter (Roth/Millipore). Virus was concentrated using 20% sucrose and ultracentrifugation at 24,000 rpm (Beckman SW28 rotor) for 2 hr at 4°C. The supernatant was discarded and 200 µl fresh FCS-free RPMI medium were added on the virus pellet and incubated for 30 min at 4°C. The pellet was resuspended and stored at –80°C or directly used for transduction. Virus titers were assessed by determination of reverse transcriptase activity (SG-PERT).

## Transduction of human T cells

$2–3 \times 10^6$ JNLA or A3.01 cells were resuspended in 1.5E11 puRT/ µl concentrated virus solution or 1 ml of nonconcentrated virus supernatant followed spin-transduction in 24-well plate format at 2300 rpm, for 1.5 hr at 37°C, RT. After transduction, the cells were incubated at 37°C, overnight. The next day the cells were transferred into a 12-well plate and 3 ml complemented RPMI medium was added and incubated overnight. Cells expressing shRNAs or the C5/C5L-mCherry constructs were transferred to fresh medium 24 hr post transduction. 48 hr later, puromycin (1.5 µg/ml) was added, and 72 hr post transduction, the medium was changed to fresh media with puromycin to accelerate cell growth. On day 4 post transduction, the cells were adjusted to the densities required according to the experimental question being addressed, with RPMI media without any selection antibiotics. Knockdown was stable in the bulk culture for up to ~1 wk post transduction. To generate stable A301 cells expressing either C5 or C5L tagged with N-terminal mCherry, the cells were FACS sorted for mCherry-expressing cells post selection with puromycin for 1 wk and then expanded in culture.

## Immunoprecipitation and RFP-Trap pulldown assay

$10 \times 10^6$ JNLA cells stably expressing mCherry alone or mCherry-ARPC5 or ARPC5L were lysed in RFP-Trap buffer (RIPA buffer: 20 mM Tris-HCl pH 7.5, 150 mM NaCl, 1 mM EDTA, 0.1% SDS, 0.5% NP-40, 0.5% Na-deoxycholate) using manual sonication at 4° (ON: 30 s; OFF:10 s, Cycles-10). BCA protein measurements were performed using the Pierce BCA protein estimation kit, followed by loading 500 µg protein per sample with 25 µl RFP-Trap bead (ChromoTek) slurry for immunoprecipitation overnight at 4° on a rotator. All other steps of the pulldown were performed according to the manufacturer's instructions (ChromoTek). Immunoblot analysis was performed with the indicated antibodies (mentioned in the reagent section).

## Immunofluorescence microscopy

As described previously (*Tsopoulidis et al., 2019*), to study the actin dynamics in JNLA cells activated on stimulatory coverslips, $1–2 \times 10^5$ cells are put on the stimulatory coverslips for 5 min at 37°C before fixing them with 3% PFA. Following permeabilization and blocking, coverslips are incubated with primary antibodies overnight at 4°C in 1% BSA (PBS). For phospho-specific targets/antibodies, all steps were done in 1× TBS. The following dilutions are used for the primary antibodies: rabbit anti-pTyr (1:100), rabbit anti-pSLP76 (1:1000), and mouse anti-mCherry (1:500). Species-specific secondary antibodies conjugated to Alexa Fluor 568/647 (1:1000) were used along with Phalloidin-Alexa Fluor 488 (1:600) for staining the F-actin. Although the nuclear lifeact reporter carries a nuclear export signal and thus also labels cytoplasmic F-actin, cortical F-actin is only labeled with low efficacy following permeabilization/fixation. An additional F-actin stain with Phalloidin is therefore required to efficiently stain and visualize cortical actin filaments. Coverslips were mounted with Mowiol (Merck Millipore) and analyzed by either epifluorescence microscopy (IX81 SIF-3 microscope and Xcellence Pro software; Olympus) or confocal microscopy (TCS SP8 microscope and LAS X software; from Leica).

## Generation of CRISPR-Cas9-based knockout cells

We designed three single-guideRNAs (sgRNA) for knocking out each of our gene of interest with the help of Synthego's publicly available CRISPR design tool (https://design.synthego.com/#/). The sgRNA sequences are also mentioned in the MDAR report. The sgRNAs were premixed with Cas9.3NLS (IDT) to create ribonucleoprotein complexes (RNPs) for faster and better editing efficiency as described earlier (*Albanese et al., 2022*). Premixed RNPs were then nucleofected (using either Amaxa 2b or Nucleofector 4D, Lonza) into the JNLA cells. JNLA cells post nucleofection are maintained in RPMI containing 10% FBS as a heterogeneous pool, followed by knockout (KO) validation in the bulk pool using immunoblotting. For single KO clone expansion, nucleofected cell pools were expanded gradually for 4–5 d until they are validated for KO. Post KO validation, the cell suspension was diluted stepwise to reach a density of 0.5 cells/50 μl to seed 50 μl/well in 96-U-bottom plates. Cells were then kept undisturbed for 1–2 wk in the incubator until we observe change in color of the media. Wells with multiple colonies growing were discarded, whereas single clonal populations were expanded gradually further until they were validated for KO using western blotting and surveyor assay, followed by initial functional characterization.

## Nuclear and cytoplasmic biochemical fractionation

JNLA cells were fractionated using the REAP method as described in *Suzuki et al., 2010*. $8 \times 10^6$ cells were harvested for each condition. The only difference we adapted is the manual sonication instead of an automated one. The number of sonication cycle varies between 10–15 (60 s ON, 10 s OFF) with the manual sonicator at 4°C or with ice. Each of the nuclear, cytoplasmic, and total cell fractions are then immunoblotted as described above. As and when necessary, immunoblots were often stripped in 1× stripping buffer, followed by blocking for 1 hr at RT and re-probing with primary antibodies ON at 4°C. The protocol we followed for stripping including the preparation of stripping buffer was adapted from Abcam's published protocol online (https://www.abcam.com/ps/pdf/protocols/stripping%20for%20reprobing.pdf).

## Single-cell RNA-seq analysis

For single-cell RNA-seq analysis of Jurkat CD4 T cell, we used data from *Zheng et al., 2017* stored at http://support.10xgenomics.com/single-cell/datasets. We obtained all data files that contained Jurkat cells and removed all non-Jurkat cells as described below.

For single-cell RNA-seq analysis of primary CD4 T cells, we obtained gene expression matrices from Gene Expression omnibus (GEO) under accession number GSE126030.

For quality control, all cells with less than 200 genes expressed were removed (CreateSeuratObject function). To reduce batch effects, we integrated the primary T cell samples (GSM3589410 resting primary T cells and GSM3589411 activated primary T cells) or (Jurkat cells, 50% Jurkat:293T and 99%:1

Jurkat:293T Cell Mixtures) via the Seurat integration procedure. In a second step, only cells with less than 5% mitochondria expression were retained. The data was normalized and scored using Seurat NormalizeData and ScaleData function and decomposed using PCA. UMAP (*McInnes et al., 2020*) embeddings were then computed using the first five (Jurkat) or eight (primary CD4 T cells) dimensions as input. Based on the UMAP embedding, further quality control was performed by removing clusters of contaminating cells (identified by differential expression) and of poor-quality cells (identified as outliers by number of genes expressed or percentage mitochondria). Subsequently, all cells that did not express T cells markers (FOXP3, CD8A, CD8B, IL2, IFNG, TNF, MIR155HG, IL4R, GZMB, MAL) were removed. To enrich for CD4 T cells, clusters expressing CD8A, CD8B, GZMB were filtered out. Primary CD4 T cell populations were annotated with conventional T cell markers (*Figure 2—figure supplement 1C and D*). Because of the low heterogeneity within the Jurkat population, all computed clusters were merged to one. Cell cycle scoring analysis was performed with cell cycle marker genes reported in *Kowalczyk et al., 2015*.

## Statistical measurements

Statistical evaluation was performed with GraphPad Prism 8 using different two-tailed tests: for two groups, unpaired *t*-test with Welch's correction (for nonequal standard deviation) was performed. Based on the experiment question asked, for multiple groups (more than two), either one-way ANOVA (Kruskal–Wallis test for nonparametric data sets) with no correction for multiple comparison was performed to keep the comparison of control to test sample as a stand-alone comparison or one-way ANOVA (for parametric data sets) with Tukey's multiple comparison was performed when multiple comparisons were analyzed. The number of independent data points always refers to biological replicates as mentioned in the legends, also indicating the technical replicates, when performed for some of the experiments. Each data point as mentioned in the figure legends represents the mean of one independent experiment with the errors calculated based on mean ± SD. Differences were considered statistically significant and denoted as follows: *$p \leq 0.0332$; **$p \leq 0.0021$; ***$p \leq 0.0002$, n.s., not significant, if $p > 0.05$.

## Material availability statement

All material generated in this study is available upon request to the corresponding authors.

# Acknowledgements

We are grateful to Nadine Tibroni and Ina Ambiel for technical assistance and Kathrin Bajak for help with manuscript preparation and submission.

We would like to acknowledge the microscopy support from the Infectious Diseases Imaging Platform (IDIP) at the Center for Integrative Infectious Disease Research, Heidelberg, Germany. This project was supported by the Deutsche Forschungsgemeinschaft (DFG, German Research Foundation) by project FA 378/20-1 to OTF. MW was supported by Cancer Research UK (CC2096), the UK Medical Research Council (CC2096), and the Wellcome Trust (CC2096) funding at the Francis Crick Institute as well as by the European Research Council (ERC) under the European Union's Horizon 2020 research and innovation program (grant agreement no. 810207 to MW). For the purpose of Open Access, the author has applied a CC BY public copyright license to any Author Accepted Manuscript version arising from this submission.

## Additional information

### Funding

| Funder | Grant reference number | Author |
|---|---|---|
| Deutsche Forschungsgemeinschaft | FA 378/20-1 | Oliver T Fackler |
| Cancer Research UK | CC2096 | Michael Way |
| Medical Research Council | CC2096 | Michael Way |
| Wellcome Trust | CC2096 | Michael Way |
| European Research Council | 810207 | Michael Way |

The funders had no role in study design, data collection and interpretation, or the decision to submit the work for publication. For the purpose of Open Access, the authors have applied a CC BY public copyright license to any Author Accepted Manuscript version arising from this submission.

### Author contributions

Lopamudra Sadhu, Data curation, Investigation, Methodology, Writing - original draft, Writing – review and editing; Nikolaos Tsopoulidis, Conceptualization, Data curation, Investigation, Writing – review and editing; Md Hasanuzzaman, Investigation; Vibor Laketa, Data curation, Investigation, Methodology, Writing – review and editing; Michael Way, Resources, Writing – review and editing; Oliver T Fackler, Conceptualization, Supervision, Funding acquisition, Writing - original draft, Writing – review and editing

### Author ORCIDs

Vibor Laketa http://orcid.org/0000-0002-9472-2738
Michael Way http://orcid.org/0000-0001-7207-2722
Oliver T Fackler http://orcid.org/0000-0003-2982-4209

### Ethics

Ethics vote not required as blood samples were obtained from fully anonymized volunteers.

### Decision letter and Author response

Decision letter https://doi.org/10.7554/eLife.82450.sa1
Author response https://doi.org/10.7554/eLife.82450.sa2

## Additional files

### Supplementary files

- MDAR checklist

### Data availability

All data are included in the manuscript. Source data files are publically available (https://doi.org/10.11588/data/YVYEO8) and described in the supplemental information section of this manuscript.

The following dataset was generated:

| Author(s) | Year | Dataset title | Dataset URL | Database and Identifier |
|---|---|---|---|---|
| Sadhu L, Tsopoulidis N, Hasanuzzaman M, Laket V, Way M, Fackler OT | 2023 | ARPC5 Isoforms and Their Regulation by Calcium-Cal 1 modulin-N-WASP Drive Distinct Arp2/3-dependent Actin Remodeling Events in CD4 T Cells [Source Data Files] | https://doi.org/10.11588/data/YVYEO8 | heiDATA, 10.11588/data/YVYEO8 |

The following previously published datasets were used:

| Author(s) | Year | Dataset title | Dataset URL | Database and Identifier |
|---|---|---|---|---|
| Zheng GXY, Terry JM, Belgrader P, Ryvkin P, Bent ZW, Wilson R, Ziraldo SB, Wheeler TD, McDermott GP, Zhu J, Gregory MT, Shuga J, Montesclaros L, Underwood JG, Masquelier DA, Nishimura SY, Schnall-Levin M, Wyatt PW, Hindson CM, Bharadwaj R, Wong A, Ness KD, Beppu LW, Deeg HJ, McFarland C, Loeb KR, Valente WJ, Ericson NG, Stevens EA, Radich JP, Mikkelsen TS, Hindson BJ, Bielas JH | 2017 | Massively parallel digital transcriptional profiling of single cells | https://www.ncbi.nlm.nih.gov/sra/SRP073767 | NCBI Sequence Read Archive, SRP073767 |
| Szabo PA, Mendes Levitin H, Miron M, Snyder ME, Senda T, Yuan J, Cheng YL, Bush EC, Dogra P, Thapa P, Farber DL, Sims PA | 2019 | A single cell reference map for human blood and tissue T cell activation | https://www.ncbi.nlm.nih.gov/geo/query/acc.cgi?acc=GSE126030 | NCBI Gene Expression Omnibus, GSE126030 |

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

# Appendix 1

## Appendix 1—key resources table

| Reagent type (species) or resource | Designation | Source or reference | Identifiers | Additional information |
|---|---|---|---|---|
| Cell line (*Homo sapiens*) | HEK293T cell line | *Tsopoulidis et al., 2019* | RRID:CVCL_0063 | |
| Cell line (*H. sapiens*) | Jurkat Tag (JTag) cells clone E6-1, stably expressing nuclear lifeact.GFP | *Tsopoulidis et al., 2019* | Source cells-Jurkat Tag; RRID:CVCL_C831 | |
| Cell line (*H. sapiens*) | CEM-derived A3.01 cell line | *Tsopoulidis et al., 2019* | RRID:CVCL_6244 | |
| Cell line (*H. sapiens*) | Raji B cell line | *Tsopoulidis et al., 2019*; *Kaw et al., 2020* | RRID:CVCL_0511 | |
| Recombinant DNA reagent | Construct pLVX-mCherry-siRES-hARPC5L (*Human*) plasmid | *Abella et al., 2016* | Kind gift from Michael Way's lab | Lentiviral construct to express the mCherry-tagged ARPC5L. |
| Recombinant DNA reagent | Construct pLVX-mCherry-siRES-hARPC5 (*Human*) plasmid | *Abella et al., 2016* | Kind gift from Michael Way's lab | Lentiviral construct to express the mCherry-tagged ARPC5. |
| Recombinant DNA reagent | Construct pLVX-puro-mCherry (*Human*) plasmid | *Abella et al., 2016* | Kind gift from Michael Way's lab | Lentiviral construct to express the mCherry |
| Recombinant DNA reagent | pLKO.1-Puro-shRNA | *Tsopoulidis et al., 2019*, Sigma-Aldrich | RRID:Addgene_10878 | Lentiviral construct to express the shRNAs |
| Antibody | Anti-human ARP3 (mouse monoclonal) | Sigma-Aldrich | Clone FMS338: Cat# A5979; RRID:AB_476749 | WB (1:10,000) |
| Antibody | Anti-human p16 ARC/ARPC5, (mouse monoclonal) | Synaptic Systems | Cat# 305011; RRID:AB_887896 | WB (1:500) |
| Antibody | Anti-human ARPC5L (rabbit polyclonal) | GeneTex | Cat# GTX120725; RRID:AB_11172404 | WB (1:1000) |
| Antibody | Anti-human ARPC1A (rabbit polyclonal) | Sigma-Aldrich | Cat# HPA004334 | WB (1:500) |
| Antibody | Anti-human ARPC1B (mouse monoclonal) | Santa Cruz Biotechnology | Cat# sc-137125; RRID:AB_2289927 | WB (1:500) |
| Antibody | Anti-human WASL (rabbit polyclonal) | Sigma-Aldrich | Cat# HPA005750; RRID:AB_1854729 | WB (1:500) |
| Antibody | Anti-human WASHC5 (rabbit polyclonal) | Sigma-Aldrich | Cat# HPA070916 | WB (1:250) |
| Antibody | Anti-human WAVE2 (mouse monoclonal) | Santa Cruz Biotechnology | Cat# sc-373889; RRID:AB_10917394 | WB (1:500) |
| Antibody | Anti-mCherry (rabbit polyclonal) | Abcam | Cat# ab167453; RRID:AB_2571870 | WB (1:1000) IF (1:500) |
| Antibody | Anti-mCherry (mouse monoclonal) | Novus | Cat# NBP1-96752SS; RRID:AB_11008969 | WB (1:1000) IF (1:500) |
| Antibody | Anti-pTyr (rabbit polyclonal) | Santa Cruz Biotechnology | Cat# sc-18182; RRID:AB_670513 | IF (1:100) |
| Antibody | Anti-human pSLP76 (rabbit polyclonal) | Abcam | Cat# ab75829; RRID:AB_2136886 | IF (1:1000) |
| Antibody | Brilliant Violet 421 anti-human TNF-α antibody (mouse monoclonal) | BioLegend | Cat# 502932; RRID:AB_10960738 | Flow cytometry (1:100) |
| Antibody | APC anti-human IL-2 antibody (rat monoclonal) | BioLegend | Cat# 500311; RRID:AB_315098 | Flow cytometry (1:100) |
| Antibody | FITC mouse anti-human CD3 antibody (mouse monoclonal) | BD Biosciences | Cat# 561802; RRID:AB_10893003 | Flow cytometry (1:100) |
| Sequence-based reagent | ARPC5_F | Primer Bank, MGH-PGA | PCR primers | TGGTGTGGATCTCCTAATGAAGT |

*Appendix 1 Continued on next page*

*Appendix 1 Continued*

| Reagent type (species) or resource | Designation | Source or reference | Identifiers | Additional information |
|---|---|---|---|---|
| Sequence-based reagent | ARPC5_R | Primer Bank, MGH-PGA | PCR primers | CACGAACAATGGACCCTACTC |
| Sequence-based reagent | ARPC5L_F | Primer Bank, MGH-PGA | PCR primers | TCTCCCGTCAACACCAAGAAT |
| Sequence-based reagent | ARPC5L_R | Primer Bank, MGH-PGA | PCR primers | GCCTGCTCAATCTCACTGCT |
| Sequence-based reagent | ARPC1A (human) | Sigma-Aldrich | shRNA target sequence | CCCTGGTGATCCTGAGAATTA |
| Sequence-based reagent | ARPC1B (human) | Sigma-Aldrich | shRNA target sequence | GCTGACCTTCATCACAGACAA |
| Sequence-based reagent | ARPC5 (human) | Sigma-Aldrich | shRNA target sequence | GTTCAATCTCTGGACAAGAAT |
| Sequence-based reagent | ARPC5L (human) | Sigma-Aldrich | shRNA target sequence | GAAAGTGCTCACAAACTTCAA |
| Sequence-based reagent | ARPC5 (Human) | Synthego | sgRNA sequences | sgRNA1: GCAGUGCUAUGUUACUGCAA sgRNA2: CAAUGCUGCCUGCCCGGUCC sgRNA3: UGACUCUUGGUGUUGAUAGG |
| Sequence-based reagent | ARPC5L (human) | Synthego | sgRNA sequences | sgRNA1: UCGUCUGCAGGAGCGAGCCC sgRNA2: ACUGCGCUGCUAUUUUCUGU sgRNA3: AUUCGUCGAUGUCCACCCGG |
| Commercial assay or kit | WesternBright Sirius Chemiluminescent Detection Kit | Advansta | Cat# K-12043-D20; RRID:SCR_013577 | ECL-based detection of proteins |
| Commercial assay or kit | RFP-Trap Magnetic Agarose | ChromoTek Proteintech | Cat# rtma-100; AB_2631363 | For immunoprecipitation of mCherry-tagged proteins |
| Peptide, recombinant protein | Alt-R S.p. Cas9 Nuclease V3 | IDT Germany | Cat# 1081059 | For CRISPR-Cas9 nucleofection reaction |
| Chemical compound, drug | PMA | Sigma-Aldrich | Cat# P1585-1MG | |
| Chemical compound, drug | Ionomycin | Sigma-Aldrich | Cat# I0634-1MG | |
| Chemical compound, drug | CK-869 ≥ 98% (HPLC) | Sigma-Aldrich | Cat# C9124 | |
| Chemical compound, drug | Aphidicolin, Ready Made Solution - 1 ml | Sigma-Aldrich | Cat# A4487 | |
| Software, algorithm | Fiji/ImageJ | Fiji/ImageJ | RRID:SCR_002285; PMID:22743772 | Image processing |
| Software, algorithm | FlowJo | BD Biosciences | RRID:SCR_008520 | Software for flow cytometry data analysis |
| Software, algorithm | Prism 8 | GraphPad | RRID:SCR_002798 | Data analysis and quantification |
| Software, algorithm | Illustrator CC | Adobe | RRID:SCR_010279 | Vector graphics and assembly |
| Software, algorithm | bioRENDER | bioRENDER (paid license) | RRID:SCR_018361 | Graphical illustrations |
| Other | 4D Nucleofector -Core+X unit | Lonza Biosciences | Cat# AAF-1003X | For nucleofection |
| Other | Spinning-disk confocal microscope | Nikon Ti PerkinElmer UltraVIEW VoX | As used in *Tsopoulidis et al., 2019* | Live-cell imaging |
| Other | SLM 2D/3D STED/RESOLFT | Abberior Instruments GmbH, Göttingen, Germany | As used in *Tsopoulidis et al., 2019* | Super-resolution microscopy |
| Other | Leica SP8 TCS DLS Confocal and SPIM | Leica Microsystems | | Confocal microscopy |
| Other | FACS Celesta | BD Biosciences | | Flow cytometry |

