## [Editor Report]

This fundamental study demonstrates that the two isoforms of the ARPC5 subunit (ARPC5 and ARPC5L) of the Arp2/3 complex have specific functions in regulating cytoplasmic and nuclear actin filament assembly in response to DNA replication stress and T cell receptor signaling in T lymphocytes. The data presented in the manuscript are convincing and of good technical quality, and the study provides interesting new insights into specific cellular roles of different Arp2/3 isoforms in T lymphocytes.

---

## [Decision Letter]

**Decision letter after peer review:**

Thank you for submitting your article "ARPC5 Isoforms and Their Regulation by Calcium-Calmodulin-N-WASP Drive Distinct Arp2/3-dependent Actin Remodeling Events in CD4 T Cells" for consideration by *eLife*. Your article has been reviewed by 3 peer reviewers, including Pekka Lappalainen as Reviewing Editor and Reviewer #3, and the evaluation has been overseen by Anna Akhmanova as the Senior Editor.

Essential revisions:

1. The imaging experiments focusing on the nuclear localization of ARPC5 and ARPC5L should be strengthened. The authors need to examine how well do the tagged proteins incorporate into native Arp2/3 complex. This could be done e.g. with native PAGE. Moreover, possible co-localization of ARPC5 and ARPC5L with N-WASP in the nucleus could be examined.

2. The authors should examine if there are any functional consequences of losing either ARPC5 or ARPC5L. This could be approached e.g. by using Jurkat cells, because the authors previously showed that nuclear actin has consequences for cytokine expression in Jurkat cells.

3. There are also some technical issues that should be addressed to strengthen the study, and these are listed in the 'specific comments' by the three reviewers.

*Reviewer #1 (Recommendations for the authors):*

Critique:

Figure 1 – not too clear how the quantifications were done.

"All data points indicate mean{plus minus} s.d values from three independent experiments with at least 40 cells analyzed per condition per experiment".

There are only 3 data points within the bar charts shown! It should be made more clear how this was quantified such as double blind, etc…

1F is the response transient? Or are these filaments stable over long time?

1G only about 25% cells respond with nuclear actin formation?

Figure 2 seems to be more of a supplemental information perhaps?

Figure 3, only 1 shRNA was used, no rescue experiments were performed. I understand the difficulty and complexity of the cellular system used but maybe the authors can comment here. Because this is a limitation for conclusions made.

Figure 4, why was PMA plus Inonomycin used here and how does that compare to the other stimuli or using them singularly?

Figure 6 and 7, how did they determine whether they actually successfully and reliably induced replication stress? This is important and should be addressed by the authors.

*Reviewer #2 (Recommendations for the authors):*

Since disruption of Arp2/3 complex will affect membrane trafficking and therefore might change the levels of surface receptors present on cells, do the authors know whether some of the differences that they observe might be due to different amounts of TCR displayed on the cell surface in response to CK689 treatment?

All of the experiments are done in Jurkat T-cells. The importance of the results would be enhanced if at least the presence of nuclear actin in response to the various signals were shown in primary T-cells.

The authors use tagged ARPC5 and APRC5L to show that both isoforms can be found in the nucleus. The results presented are mostly convincing, but it isn't clear that the tagged constructs behave as endogenous proteins. Can they assess the percentage of incorporation of these into the Arp2/3 complex using a system such as native PAGE gels? This could support (or not) whether the whole Arp2/3 complex is in the nucleus with the attached subunit or just the free subunit that they expressed.

The finding that different nucleation promotion factors appear to be required for different signal-induced actin events is potentially interesting, but unexplained. The authors hint that this might be due to effects on expression levels of ARPC5A being tied somehow to N-WASP. They also find some changes in expression of ARPC1 isoforms in response to knockdown/out of ARPC5 isoforms. This is complex and it is hard to tell how direct the effects might be.

The authors made the interesting connection between nuclear calmodulin and actin assembly (Figure 6), suggesting some connection with N-WASP. It would be of great interest to know how nuclear calmodulin connects with N-WASP.

It is mentioned that the function of nuclear actin assembly might be to change gene expression. The ARPC5/5L CRISPR cells might provide an ideal opportunity to explore this further and demonstrate a biological role for these differences that could have a physiological meaning. This could add interest to the paper in terms of immunology and T-cell function.

In summary, this study highlights some interesting observations that suggest that ARP2/3 complex subunit isoforms might have signal-dependent differences in function in CD4^+^ T-cells. However, the mechanisms by which this works are not yet clear and in some cases, the effects are not explained- e.g. whether they are due to gene expression, direct protein interactions or a chain of signaling events. The study would make a larger impact in the field if some of these questions could be addressed.

*Reviewer #3 (Recommendations for the authors):*

1. The Western blots of NPF knockout cells in Figure 7A look confusing. This is the case especially with the N-WASP knockout cells, in which virtually all unspecific bands detected in the control cell extract are not visible in the knockout cell sample. This indicates that the authors have not loaded identical 'concentrations' of control cell and knockout cell lysates on the gels, or that all 'unspecific' bands visible in the control sample correspond to degradation products of N-WASP. This should be clarified, or alternatively, the NFP data should be deleted from the manuscript.

2. The co-localization studies presented in Figure 5 appear somewhat preliminary. These would be much more informative if the authors could also examine the possible co-localization of ARPC5 and ARPC5L with their NPFs in different conditions. If technically possible, it would be also interesting to know if the ARPC5 and ARPC5L puncta co-localize with each other in the nuclei.

3. The scale bars should be precisely defined in all figures (and panels). As an example, the authors state in the legend to Figure S1 that the scale bar is 7 um, but this is certainly not the case in all panels of the figure.

---

## [Author Response]

Essential revisions:1. The imaging experiments focusing on the nuclear localization of ARPC5 and ARPC5L should be strengthened. The authors need to examine how well do the tagged proteins incorporate into native Arp2/3 complex. This could be done e.g. with native PAGE. Moreover, possible co-localization of ARPC5 and ARPC5L with N-WASP in the nucleus could be examined.

The Way lab had previously demonstrated in HeLa cells by co-immunoprecipitation that ectopically expressed C5 and C5L isoforms are assembled into Arp2/3 complexes with similar efficacy (Abella JV et al., 2016). We performed analogous experiments in Jurkat CD4 T cells transiently expressing C5.mCherry or C5L.mCherry. The results, now shown as figure S3D in the revised manuscript, reveal that both ectopically expressed C5 isoforms are incorporated into endogenous Arp2/3 complexes with comparable efficacy.

We also attempted to address whether ARPC5 and ARPC5L colocalize with N-WASP in the nucleus of CD4 T cells. Since we lack antibodies that reliably discriminate between both C5 isoforms in immunofluorescence and the various anti N-WASP antibodies we tested all gave marked background signals in CD4 T cells, this had to be approached by ectopic expression of ARPC5/C5L as well as N-WASP. We attempted this by co-transfection as well as by simultaneous as well as consecutive lentiviral transduction, however the frequency of cells with detectable co-expression of both factors was too low (below 2%) to allow robust co-localization analysis. Overexpression of C5 isoforms with the NPF thus appears to be toxic in CD4 T cells, which prevented us from performing this relevant analysis. This is now discussed in the revised version of the manuscript.

2. The authors should examine if there are any functional consequences of losing either ARPC5 or ARPC5L. This could be approached e.g. by using Jurkat cells, because the authors previously showed that nuclear actin has consequences for cytokine expression in Jurkat cells.

As requested, we assessed cytokine production in response to T cell activation in a CD4 T cell line that responds to PMA+Ionomycin stimulation by the formation of nuclear actin filaments (A3.01 cells, see Tsopoulidis et al., 2019) and the production of TNF-a as well as IL-2. Knockdown experiments reveal that both, ARPC5L and ARPC5, are required for full cytokine production (new figure 4I-J, S5D-E). Thus, cytoplasmic as well as nuclear actin dynamics and hence also both ARPC5 isoforms contribute to cytokine production of CD4 T cells in response to T cell activation.

3. There are also some technical issues that should be addressed to strengthen the study, and these are listed in the 'specific comments' by the three reviewers.

see detailed response to the specific points raised by the individual reviewers.

Reviewer #1 (Recommendations for the authors):Critique:Figure 1 – not too clear how the quantifications were done."All data points indicate mean{plus minus} s.d values from three independent experiments with at least 40 cells analyzed per condition per experiment".There are only 3 data points within the bar charts shown! It should be made more clear how this was quantified such as double blind, etc…

We apologize for this confusion. As mentioned in the figures legend, each data point represents the mean of one each independent experiment in which at least 40 cells were analyzed for each condition to yield this mean value. This is now explained better in the revised manuscript.

1F is the response transient? Or are these filaments stable over long time?

The nuclear actin filament network induced in response to T cell activation is transient. As we characterized in our previous study, filament formation initiates within seconds of stimulatory contacts and the network is typically dissolved after approximately 5 min in the majority of cells (Tsoupoulidis et al. 2019). Formation of the nuclear network precedes actin polymerization in the cytoplasm, which is typically initiated 2 min post stimulatory contact and the resulting F-actin structure (circumferential F-actin ring) persists over 10-20 min. The transient nature of the nuclear F-actin network and the different timings of cytoplasmic vs nuclear F-actin structures is now emphasized in the revised text.

1G only about 25% cells respond with nuclear actin formation?

The reviewer is correct, nuclear F-actin is only detected in a fraction of stimulated cells. The percentage of cells responding with nuclear actin formation slightly varies between different types of stimulation but ranges between 22-30% between PMA+ionomycin stimulation and stimulatory coverslips for in Jurkat cells. While we cannot exclude that the sensitivity of our visual detection of nuclear F-actin represents a limitation, it is noteworthy that responses of CD4 T cells cultures are known to be heterogeneous and e.g. not all cells express cytokines in response to activation (Figure S6D-E). As a first possible explanation for this heterogeneity, we mined available single cell RNASeq data for expression of ARPC5 and ARPC5L. This analysis revealed that distinct subfractions of CD4 T cell lines but also primary CD4 T cells express one or the other subunit isoform and that the frequency with which cells display nuclear actin polymerization in response to T cell activation roughly matches that of cells that express ARPC5L. This suggests that the ARPC5 isoform expression status governs the response only individual CD4 T cells to activation. These results are now included as Figures 2B-C and S2C-E in the revised manuscript.

Figure 2 seems to be more of a supplemental information perhaps?

We added the new single cell RNASeq data to this figure to address the important aspect raised by the reviewer above and hence kept the figure in the main body of the manuscript.

Figure 3, only 1 shRNA was used, no rescue experiments were performed. I understand the difficulty and complexity of the cellular system used but maybe the authors can comment here. Because this is a limitation for conclusions made.

We fully agree that the shRNA data alone are not conclusive. Since knock down efficiencies achieved by shRNA were moderate, we moved on to knock out the respective genes, which resulted in more robust protein depletion and is used for most of the reminder of the study. We therefore also conducted the rescue experiments in knock out cells. Since the results from shRNA and knock out experiments are consistent, we did not carry out rescue experiments in the shRNA treated cells.

Figure 4, why was PMA plus Inonomycin used here and how does that compare to the other stimuli or using them singularly?

As shown in Figure 1, stimulation by anti-CD3/28 or PMA/Ionomycin results in the induction of comparable nuclear actin dynamics, but PMA/Iono does not induce cytoplasmic actin polymerization because it bypasses proximal TCR signaling and activates the cascade downstream of plasma membrane associated factors, thus it represents a useful tool to activate T cells by circumventing the need for actin polymerization driven receptor and tyrosine kinase clustering at the plasma membrane. It is therefore used when we want to focus our analysis on the immediate signaling that impacts nuclear events. We saw previously (Tsopoulidis et al., 2019) that ionomycin alone is sufficient to trigger nuclear F-actin formation but that the combination of PMA/Ionomycin is slightly more efficient. We therefore use this combination in our experiments here and explain this in more detail in the revised text.

Figure 6 and 7, how did they determine whether they actually successfully and reliably induced replication stress? This is important and should be addressed by the authors.

As shown now in Suppl.Figure 8B of our revised manuscript, the efficacy of replication stress induction by Aphidicolin was validated by the induction of CHK-1 phosphorylation.

Reviewer #2 (Recommendations for the authors):Since disruption of Arp2/3 complex will affect membrane trafficking and therefore might change the levels of surface receptors present on cells, do the authors know whether some of the differences that they observe might be due to different amounts of TCR displayed on the cell surface in response to CK689 treatment?

We determined this as requested. The results, now shown as revised Suppl.Figure 1, reveal that CD3 cell surface levels are not significantly affected by CK869 treatment within the timeframe of our imaging experiments.

All of the experiments are done in Jurkat T-cells. The importance of the results would be enhanced if at least the presence of nuclear actin in response to the various signals were shown in primary T-cells.

We previously documented that primary human CD4 T cells form nuclear actin filaments in response to T cell activation signals (Tsopoulidis et al., 2019) and emphasize this more in the revised text. Building on this finding as well as our new technology to achieve activation neutral gene editing in resting CD4 T cells (Albanese et al., 2022 Nature Methods), we extensively tried to conduct as many experiments in primary CD4 T cells as possible. However, live visualization of nuclear F-actin formation relies on prior transduction of the nuclear lifeact.GFP reporter constructs, which is only efficient in activated CD4 T cells. We therefore had to activate the cells, transduce, wait until the activation state returned to baseline, and then re-activate to visualize the formation of nuclear F-actin. This procedure was associated with significant cytotoxicity and it was impossible to distinguish between TCR-specific and unspecific nuclear F-actin event. Combining this approach with knock down or knock out procedures, as would be required to conduct functional experiments in primary cells, did not yield any viable cells to study. We therefore had to focus our experiments on cell line models.

The authors use tagged ARPC5 and APRC5L to show that both isoforms can be found in the nucleus. The results presented are mostly convincing, but it isn't clear that the tagged constructs behave as endogenous proteins. Can they assess the percentage of incorporation of these into the Arp2/3 complex using a system such as native PAGE gels? This could support (or not) whether the whole Arp2/3 complex is in the nucleus with the attached subunit or just the free subunit that they expressed.

Please see response to the first part of essential revisions, point 1.

The finding that different nucleation promotion factors appear to be required for different signal-induced actin events is potentially interesting, but unexplained. The authors hint that this might be due to effects on expression levels of ARPC5A being tied somehow to N-WASP. They also find some changes in expression of ARPC1 isoforms in response to knockdown/out of ARPC5 isoforms. This is complex and it is hard to tell how direct the effects might be.

We agree and now emphasize that our data does not allow to distinguish whether the NPF -c5/c5L relationship is direct or indirect in the revised discussion.

The authors made the interesting connection between nuclear calmodulin and actin assembly (Figure 6), suggesting some connection with N-WASP. It would be of great interest to know how nuclear calmodulin connects with N-WASP.

We had very briefly mentioned in the previous version of the manuscript that reports in the literature suggest that calcium-calmodulin can activate N-Wasp by direct binding or via activation of the N-Wasp regulator IQGAP and significantly expanded the discussion of this aspect.

It is mentioned that the function of nuclear actin assembly might be to change gene expression. The ARPC5/5L CRISPR cells might provide an ideal opportunity to explore this further and demonstrate a biological role for these differences that could have a physiological meaning. This could add interest to the paper in terms of immunology and T-cell function.In summary, this study highlights some interesting observations that suggest that ARP2/3 complex subunit isoforms might have signal-dependent differences in function in CD4^+^ T-cells. However, the mechanisms by which this works are not yet clear and in some cases, the effects are not explained- e.g. whether they are due to gene expression, direct protein interactions or a chain of signaling events. The study would make a larger impact in the field if some of these questions could be addressed.

We agree that the discovery that cytoplasmic and nuclear actin polymerization events can be mediated by differentially configurated Arp2/3 complex opens avenues to gain more insight into the functional role of nuclear actin dynamics. This certainly includes the question how nuclear F-actin governs selective gene expression programs. The analysis of available RNASeq data sets revealed that indeed, Arpc5L expression seems to be increased in cytokine expressing effector CD4 T cells relative to non-cytokine expressing cells (see revised Figure 2, Figure S2). While this suggests that Arpc5L isoforms might play a role in shaping T cell identity and the immune response in general, addressing this experimentally is very complicated since the microscopic analysis of nuclear actin dynamics as immediate response to T cell activation needs to be coupled to cytokine detection in the very same cell many hours later. We started to explore the use of advanced microfluidics systems in which microscopy can be coupled to morphotype-based selection of individual cells but are far from having a robust system for such analysis available. We therefore feel that exploiting the subunit isoform selectivity identified here as tools for mechanistic dissection represents an ambitious new study. This future perspective is now mentioned in the revised text.

Reviewer #3 (Recommendations for the authors):1. The Western blots of NPF knockout cells in Figure 7A look confusing. This is the case especially with the N-WASP knockout cells, in which virtually all unspecific bands detected in the control cell extract are not visible in the knockout cell sample. This indicates that the authors have not loaded identical 'concentrations' of control cell and knockout cell lysates on the gels, or that all 'unspecific' bands visible in the control sample correspond to degradation products of N-WASP. This should be clarified, or alternatively, the NFP data should be deleted from the manuscript.

We agree that unspecific bands were not labelled consistently on the previous version of this figure. A re-analysis of these Western blots, including analysis with alternative anti-N-WASP antibodies (none of which recognized less unspecific signals in CD4 T cell lysates than the one use here) and quantification relative to the GAPDH loading control, confirmed that the two bands in the 72kDa range are specific while the lower molecular weight signals are unspecific. We adjusted the labeling accordingly. As now shown as supplementary figure 9D and E, the role of N-Wasp for nuclear actin polymerization in response to T cell activation, was also confirmed by pharmacological inhibition.

2. The co-localization studies presented in Figure 5 appear somewhat preliminary. These would be much more informative if the authors could also examine the possible co-localization of ARPC5 and ARPC5L with their NPFs in different conditions. If technically possible, it would be also interesting to know if the ARPC5 and ARPC5L puncta co-localize with each other in the nuclei.

Please see our response to the second part of point 1 of the essential revisions.

3. The scale bars should be precisely defined in all figures (and panels). As an example, the authors state in the legend to Figure S1 that the scale bar is 7 um, but this is certainly not the case in all panels of the figure.

We have corrected this throughout in our revised manuscript.